# Gold nanoclusters-assisted delivery of *NGF* siRNA for effective treatment of pancreatic cancer

Yifeng Lei[1,†], Lixue Tang[1], Yangzhouyun Xie[1], Yunlei Xianyu[1], Lingmin Zhang[1], Peng Wang[1], Yoh Hamada[2], Kai Jiang[3], Wenfu Zheng[1] & Xingyu Jiang[1,4]

Pancreatic cancer is one of the deadliest human cancers, whose progression is highly dependent on the nervous microenvironment. The suppression of gene expression of nerve growth factor (NGF) may have great potential in pancreatic cancer treatment. Here we show that gold nanocluster-assisted delivery of siRNA of *NGF* (GNC–siRNA) allows efficient *NGF* gene silencing and pancreatic cancer treatment. The GNC–siRNA complex increases the stability of siRNA in serum, prolongs the circulation lifetime of siRNA in blood and enhances the cellular uptake and tumour accumulation of siRNA. The GNC–siRNA complex potently downregulates the NGF expression in Panc-1 cells and in pancreatic tumours, and effectively inhibits the tumour progression in three pancreatic tumour models (subcutaneous model, orthotopic model and patient-derived xenograft model) without adverse effects. Our study constitutes a straightforward but effective approach to inhibit pancreatic cancer via *NGF* knockdown, suggesting a promising therapeutic direction for pancreatic cancer.

[1] Beijing Engineering Research Center for BioNanotechnology and CAS Key Laboratory for Biological Effects of Nanomaterials and Nanosafety, CAS Center for Excellence in Nanoscience, National Center for NanoScience and Technology, Beijing 100190, China. [2] Department of Nano-Medical Science, Graduate School of Medicine, Tohoku University, Sendai 980-8575, Japan. [3] Department of Hepatobiliary Surgery, Chinese PLA General Hospital, Beijing 100853, China. [4] University of Chinese Academy of Sciences, Beijing 100049, China. † Present address: Cross-Disciplinary Institute of Engineering Sciences, School of Power and Mechanical Engineering, Wuhan University, Wuhan 430072, China. Correspondence and requests for materials should be addressed to X.J. (email: xingyujiang@nanoctr.cn) or to W.Z. (email: zhengwf@nanoctr.cn).

Pancreatic cancer is one of the deadliest human cancers, with a 5-year survival of <5% (ref. 1). Multimodal treatment regimens combining the first-line chemotherapeutic drugs have only increased median patient survival from 5.0 to 7.2 months[1]. Thus, new therapeutic approaches are urgently needed for the treatment of this lethal disease.

Recently, nervous microenvironment has been recognized as a novel niche for cancer progression and metastasis[2–5]. In particular, nervous microenvironment has a crucial impact during the growth and metastasis of pancreatic cancer[6,7]. Perineural invasion is a prominent pathologic feature of pancreatic cancer[6], which is considered as the foremost reason for the high tumour recurrence, severe neuropathic pain and poor patient survival of pancreatic cancer[6]. Increased neurite densities are frequent pathologic features of pancreatic cancer[8]. Pancreatic tumours actively promote the growth of neurites and stimulate neurogenesis via the expression of neurotrophic factors such as nerve growth factors (NGFs) and brain-derived growth factors[9]. Among them, NGFs appear to be the most critical regulator of the tumour-induced neurogenesis. The expressions of *NGF* transcript and protein in pancreatic cancer cells and in human pancreatic tumours were reported previously[10–12]. NGF, together with its receptors, is expressed in pancreatic tumours, which contribute to their survival, proliferation, invasion and metastasis[12–15]. These observations suggest that anti-neurogenic therapy by targeting *NGF* gene has great potential for pancreatic cancer treatment.

For the intervention of gene expression, small interfering RNA (siRNA) is a short double-stranded RNA, which can achieve sequence-specific gene silencing of the complementary messenger RNA (mRNA), inducing the degradation of mRNA and inhibiting the production of target protein[16,17]. The siRNA-based therapy has emerged as a promising strategy to target multiple diseases[18]. However, the efficiency of gene silencing by naked siRNA is very low, because the naked siRNA molecules are rapidly degraded by nucleases in the bloodstream and experienced rapid renal clearance in the body[19,20]. Furthermore, the large size and negative charge of siRNA hamper its penetration across the cell membrane and prevent its intracellular accumulation[19,20]. Thus, efficient delivery is a key issue for bringing siRNA to the targeted cells and tissues.

Various materials have been developed for the efficient delivery of siRNA, including lipids, polymers, dendrimers, polymeric micelles and metallic core nanoparticles[21–23]. Gold nanomaterials, in particular, serve as attractive materials for nucleic acid delivery[24,25], due to their advantages, including tunable sizes and surface properties, and multiple functional capabilities[26–29]. Gold nanoparticle (GNP)-based oligonucleotide delivery exhibited attractive biological properties and induced effective gene knockdown in cells and tissues without apparent cellular toxicity and off-target effects[30–32]. Recently, novel fluorescent gold nanoclusters (GNCs) were developed with one-step reaction in our labs. Unlike the most popular GNPs (which do not fluoresce), fluorescent GNCs with sizes smaller than 3 nm comprise a specific type of gold nanomaterials, as they have fluorescence in the visible to near-infrared region[33,34].

Herein, we developed GNCs for efficient delivery of *NGF* siRNA (GNC–siRNA) to silence *NGF* gene in pancreatic cancer, aiming to inhibit pancreatic cancer progression. Our results showed that the GNC–siRNA complex increased the stability of siRNA in serum, prolonged the circulation lifetime of siRNA in blood and enhanced the cellular uptake and tumour accumulation of siRNA. The GNC–siRNA complex potently knocked down the NGF expression in pancreatic cells and in pancreatic tumours, and effectively suppressed the pancreatic tumour progression via *NGF* knockdown. Together, our study constituted a straightforward but very effective approach to inhibit pancreatic tumours, suggesting a novel therapeutic direction for pancreatic cancer.

## Results

**NGF expression and neurites in human pancreatic cancer**. To assess the relevance of innervation in human pancreatic cancer, we analysed the NGF expression and neurite distribution in human pancreatic tumours from pancreatic cancer patients. The statistical analysis of western blotting indicated a significantly higher expression of NGF in pancreatic tumour tissues than in normal pancreas ($P < 0.01$, $n = 4$; Supplementary Fig. 1a). Immunohistochemistry (IHC) and immunofluorescence (IF) assays of NGF confirmed that the pancreatic tumours exhibited significantly higher NGF immunoreactivity compared to normal pancreas (Supplementary Fig. 1b). Neurofilament-specific staining and reconstruction revealed more abundant neurites within pancreatic tumour tissues compared to the normal pancreas tissues (Supplementary Fig. 1c). Together, the analysis of the biopsies revealed the widespread expression of NGF protein and abundant neurites in human pancreatic tumours, suggesting that targeting NGF is promising for pancreatic cancer treatment.

***In vitro* screening of siRNA sequences and nanocarriers**. We screened the *NGF* siRNA sequences for the greatest *NGF* knockdown. We designed different sequences of *NGF* siRNA (Supplementary Table 1), and used a conventional transfection agent Lipofectamine 2000 to screen the most effective gene sequence. Of the screened sequences (Supplementary Tables 1 and 2), *NGF* siRNA-#2 resulted in the most effective knockdown of *NGF* mRNA (Supplementary Fig. 2), thus it was used to synthesize the nanomaterials–siRNA conjugates.

Different nanomaterials were conjugated with *NGF* siRNA-#2 (referred to Supplementary Methods for preparation). Among various nanomaterials, GNCs had the strongest capacity to conjugate siRNA (226 µmol siRNA per g GNCs; Supplementary Table 3) most likely due to the smaller size and larger specific surface area of GNCs (Supplementary Fig. 3). We screened the nanomaterials–siRNA conjugates for their efficiency in *NGF* knockdown. Of the screened conjugates, the GNC–siRNA complex resulted in the greatest knockdown of *NGF* mRNA (Supplementary Fig. 4), which was used for the therapeutic studies to follow.

**Synthesis and characterization of GNC–siRNA complex**. We synthesized the GNCs from one-step reduction of $Au^{3+}$ in the presence of glutathione (GSH) and oligoarginine ($CR_9$; Fig. 1a,b). The obtained GNCs had a light-green appearance in solution (Fig. 1d), with an excitation and emission peak at 430 and 596 nm, respectively (Fig. 1e). The GNCs had a positive surface charge of $19.9 \pm 0.8$ mV (Fig. 1f), due to the surface GSH and oligoarginine that carry amine-derived positive charges (Fig. 1b). The prepared GNCs had a well-defined core structure (Fig. 1g) and the diameter was $2.6 \pm 0.5$ nm (Fig. 1h).

The *NGF* siRNA adsorbed onto cationic GNCs via electrostatic interaction (Fig. 1c). The concentration ratio of 3 µg ml$^{-1}$ siRNA to 1 µg ml$^{-1}$ GNCs was used for *NGF* siRNA delivery (Supplementary Fig. 5). Accordingly, the GNC–siRNA had a loading capacity of 226 µmol siRNA per g GNCs (Supplementary Table 3). After binding of siRNA (Supplementary Fig. 6), the GNC–siRNA complex had a diameter of $16.6 \pm 3.0$ nm (Fig. 1i,j), which contained between 4 and 16 GNCs in the complex (Supplementary Fig. 7). Atomic force microscope also revealed that the GNC–siRNA complex had a higher height ($25.6 \pm 2.4$ nm) than GNCs ($2.2 \pm 0.3$ nm; Supplementary

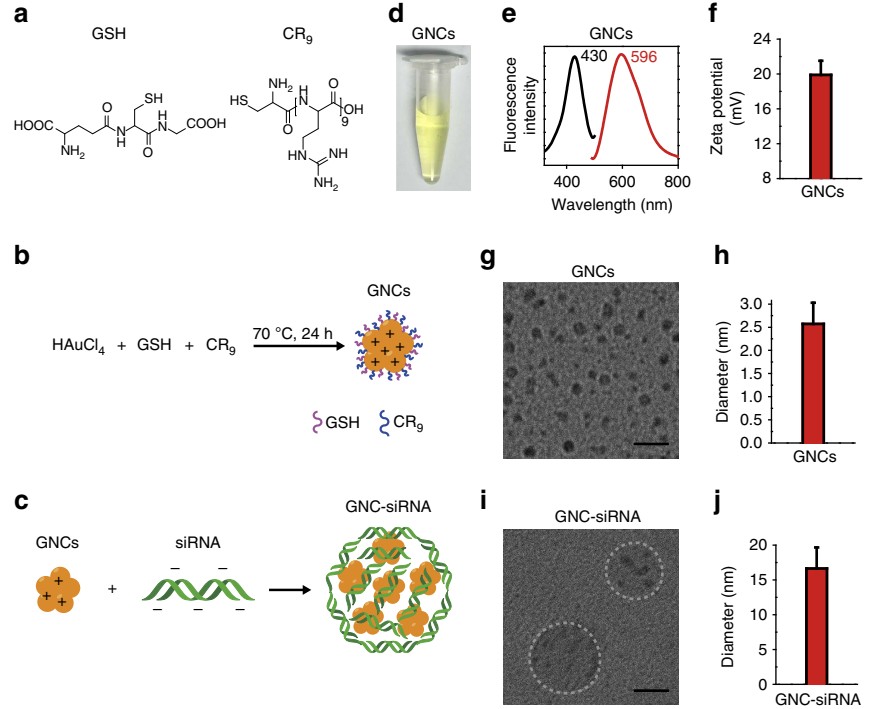

**Figure 1 | Preparation and characteristics of GNC–siRNA complex.** (**a**) Molecular structure of GSH and CR$_9$. (**b**) Scheme of the preparation of positively charged GNCs. (**c**) Scheme of the preparation of GNC–siRNA complex. The negatively charged siRNA was condensed onto cationic GNCs by electrostatic interaction. (**d**) The appearance of GNC solution. (**e**) Excitation and emission spectrum of GNCs. (**f**) Surface charge of GNCs. Mean ± s.d. ($n = 16$). (**g**) CryoTEM images of GNCs. (**h**) Diameter of the prepared GNCs. Mean ± s.d. ($n = 120$). (**i**) CryoTEM images of the GNC–siRNA complex. (**j**) Diameter of the GNC–siRNA complex. Mean ± s.d. ($n = 30$). Scale bars, 10 nm.

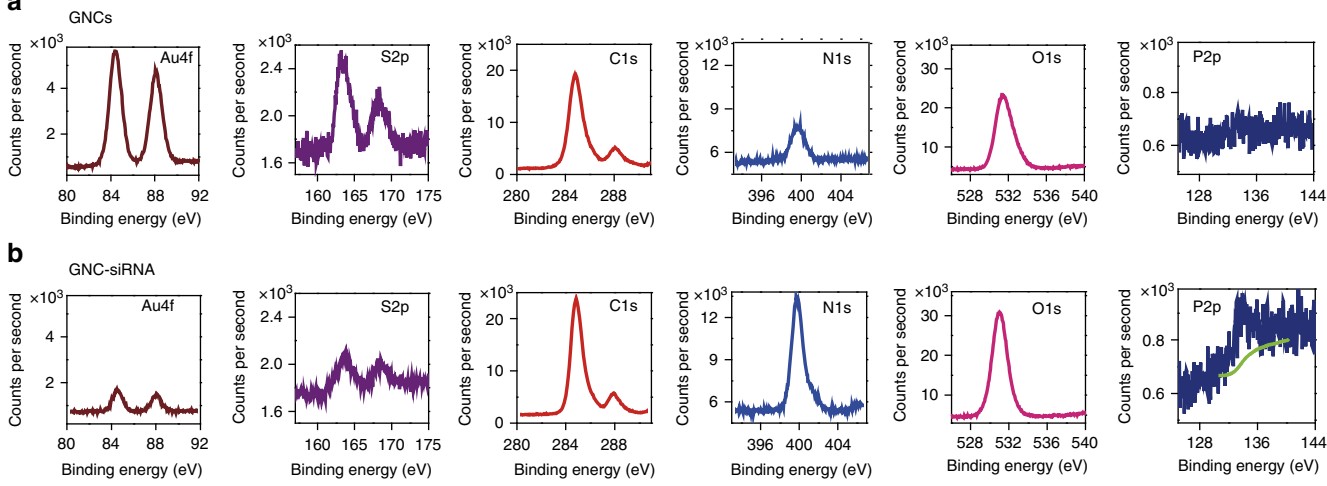

**Figure 2 | XPS analysis of GNCs and GNC–siRNA complex.** XPS spectra for the chemical element of gold (Au), sulfur (S), carbon (C), nitrogen (N), oxygen (O) and phosphor (P), respectively, on the surface of GNCs (**a**) and on the surface of GNC–siRNA complex (**b**).

Fig. 8). The GNC–siRNA complex had a hydrodynamic diameter of $70.2 \pm 8.1$ nm by dynamic light scattering measurement, and remained well dispersed in water solution with a polydispersity index of $0.224 \pm 0.016$ (Supplementary Table 4).

We confirmed the binding of siRNA onto GNCs by monitoring the changes in the atomic composition with X-ray photoelectron spectroscopy (XPS) (Fig. 2). After conjugation of siRNA onto GNCs, the surface Au and S contents of the GNC–siRNA decreased, whereas the C, N and O contents of the GNC–siRNA slightly increased compared to the GNCs (Fig. 2a,b), and the GNC–siRNA exhibited substantial surface P atom (0.54%;

Fig. 2b), due to the presence of nucleotides and phosphates of siRNA molecules (Supplementary Fig. 9 and Supplementary Table 5). These results proved successful conjugation of siRNA onto GNCs.

We investigated the effect of GNC–siRNA complex on the stability of siRNA to serum nuclease with electrophoretic mobility shift assay (Fig. 3a). At 0 min, the electrophoretic mobility of GNC–siRNA complex was less than that of the free siRNA (Fig. 3a), indicating the GNC–siRNA can form complexes that were stable against the electrophoretic force during the electrophoresis. Free siRNA and GNC–siRNA (100 nM equivalent) were

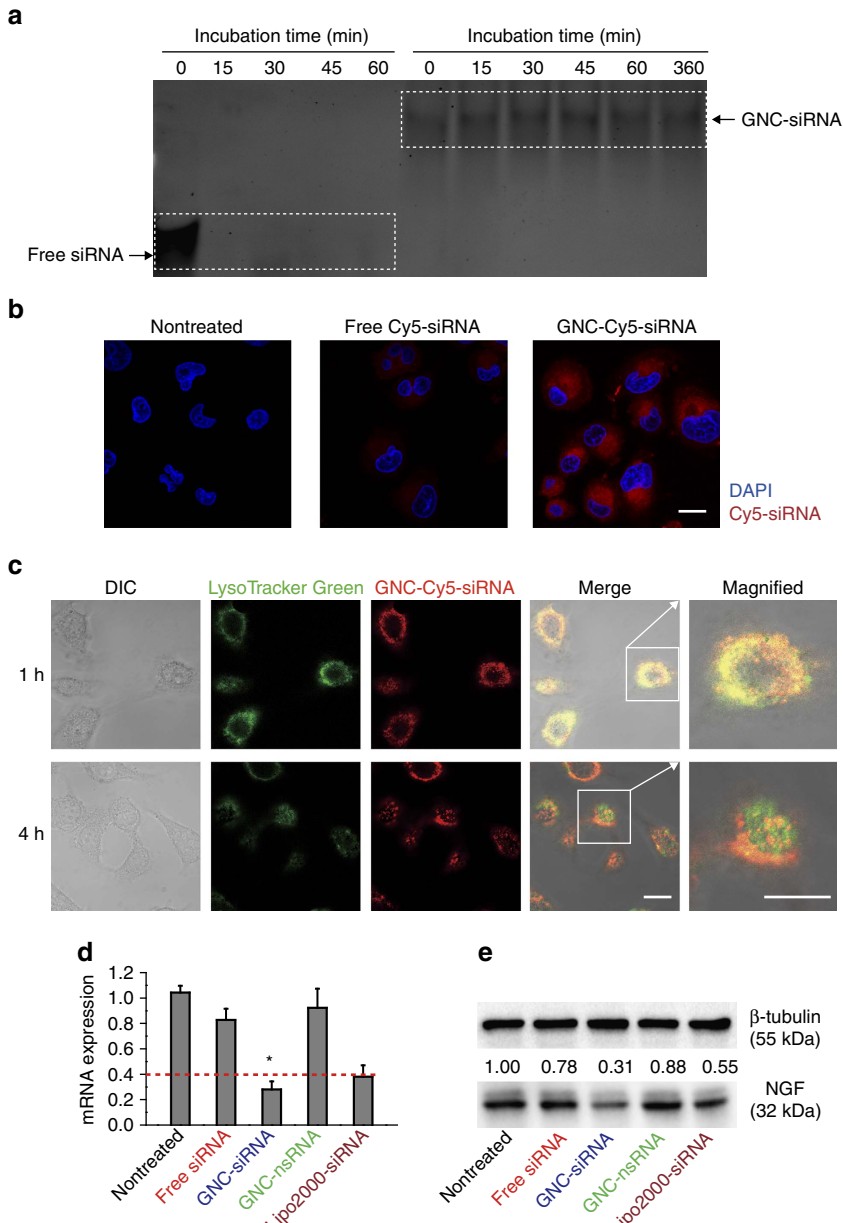

**Figure 3 | Characterization of GNC–siRNA *in vitro*.** (**a**) Protection of siRNA against serum nucleases. Free siRNA and GNC–siRNA (100 nM siRNA) were incubated within 10% serum for multiple time points, and analysed by polyacrylamide gel electrophoresis. (**b**) Cellular uptake of free siRNA and GNC–siRNA into Panc-1 cells. siRNA was labelled with Cy5 dye (Cy5-siRNA), the Panc-1 cells were incubated with various Cy5-siRNA formulations for 1 h and observed by confocal microscope with a 633 nm laser excitation. Nuclei were counterstained with 4,6-diamidino-2-phenylindole (DAPI; blue). Scale bars, 20 μm. (**c**) Lysosomal escape of GNC–siRNA in Panc-1 cells. The lysosomes of cells were stained with LysoTracker Green for 1 h, and the Panc-1 cells were treated with GNC-Cy5-siRNA for 1 h. The cells were observed by confocal microscope over different time points. Scale bars, 20 μm. (**d**) Expression level of *NGF* mRNA in Panc-1 cells analysed by RT–PCR, the dotted line referred to the expression level of *NGF* mRNA in Panc-1 cells transfected with commercially available Lipofectamine 2000 transfection agent (Lipo2000-siRNA), which served as a positive control. Mean ± s.d. (*n* = 3). *P < 0.01 compared with the nontreated control; Student's *t*-test. (**e**) Expression level of NGF protein in Panc-1 cells evaluated by western blotting. GNC binding with nsRNA was labelled as GNC–nsRNA and served as control siRNA.

incubated at 37 °C in the presence of 10% serum for multiple time points (Fig. 3a). GNC–siRNA retarded siRNA degradation in serum condition, and the intensity of siRNA remained constant over a period of 6 h incubation, indicating that GNCs protect the siRNA in the GNC–siRNA formulation (Fig. 3a). In contrast, free siRNA rapidly degraded in serum condition, and no free siRNA was detected on the gel (Fig. 3a). This result indicated that GNC–siRNA complex effectively protected the siRNA against serum nuclease degradation.

**Cellular uptake and lysosomal escape of GNC–siRNA.** Two prerequisites for efficient siRNA-mediated gene silencing are high cellular uptake of siRNA and successful release of siRNA into the cytoplasm[19]. To study the cellular uptake of siRNA, we examined the internalization of different siRNA formulations into Panc-1 cells. For visualization, *NGF* siRNA was labelled with a Cy5 dye (excitation/emission at 649/670 nm) at the 5′-end of the sense strand (Cy5-siRNA). Different Cy5-siRNA formulations were incubated with Panc-1 cells in the presence of serum for 1 h.

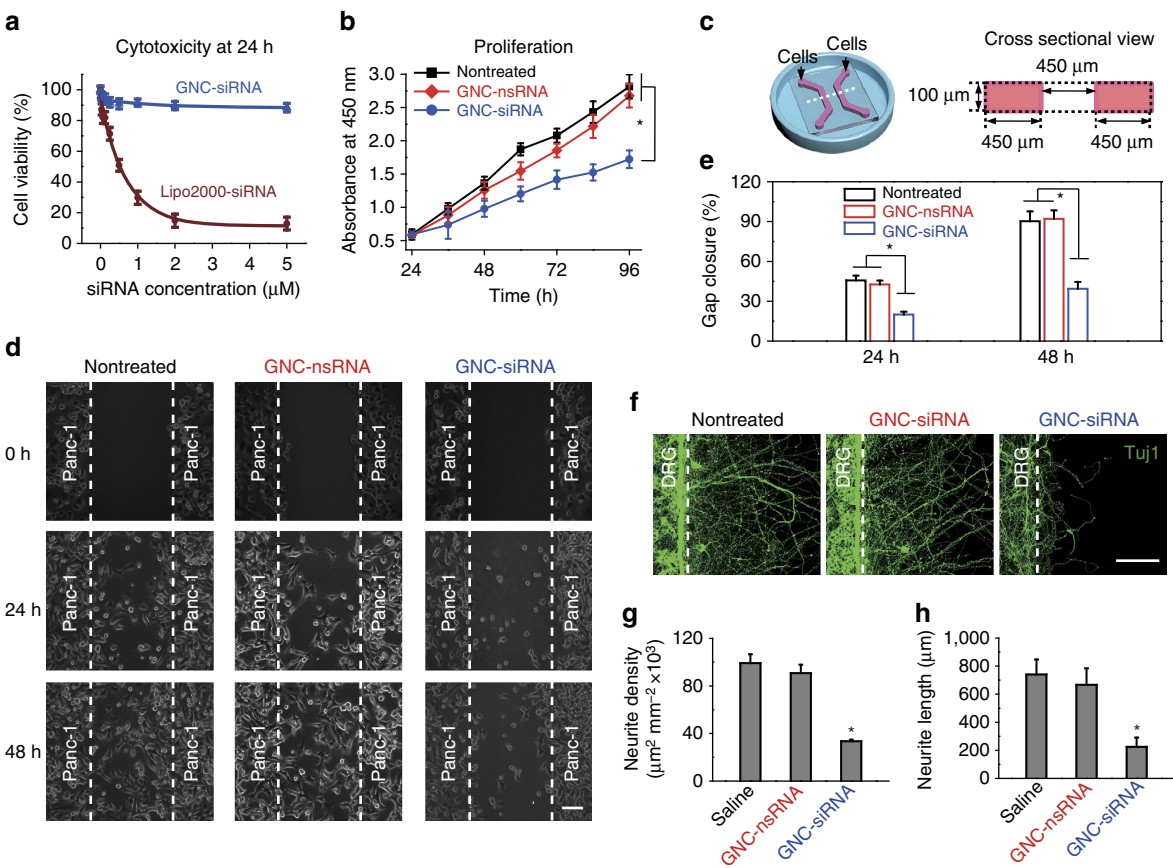

**Figure 4 | The effect of GNC–siRNA complex on cells _in vitro_.** (**a**) Viability of Panc-1 cells after 24 h incubation of GNC–siRNA versus the identical siRNA concentration coupled with Lipofectamine 2000 transfection agent (Lipo2000-siRNA). Mean ± s.d. ($n = 3$). (**b**) Cells were pre-incubated with GNC–siRNA or GNC–nsRNA (100 nM siRNA equivalent) for 24 h. Then the proliferation of Panc-1 cells within 3 days was evaluated by CCK-8 assay. Mean ± s.d. ($n = 3$). (**c**) Scheme of the microfluidic chip for cell co-culture, the right panel is the cross-sectional view of the chip. (**d**) On-chip migration assay of Panc-1 cells, which were pretreated with GNC–siRNA or GNC–nsRNA for 48 h (100 nM siRNA equivalents). The Panc-1 cells were seeded and adhered in the channels for 6 h, then the polydimethylsiloxane (PDMS) cover were peeled off ($t = 0$ h) and the migration of Panc-1 cells was monitored by microscope. Scale bars, 100 μm. (**e**) Extent of gap closure (%) at 24 and 48 h, respectively. The quantification was conducted from at least 10 fields for each condition (mean ± s.d.). (**f**) On-chip co-culture of DRG neurons and Panc-1 cells to assess the neurite sprouting. The Panc-1 cells were pretreated with GNC–siRNA or GNC–nsRNA for 48 h (100 nM siRNA). The DRG neurons and Panc-1 cells were seeded into the left and right channels of the chips, and adhered for 6 h before the removal of the cover. The neurite sprouting of DRG neurons from the neuron channel was evaluated. Neurites were stained with Tuj1 antibody and observed by confocal microscope. Scale bars, 100 μm. (**g**) Density of DRG neurite sprouting in the microfluidic chips. Mean ± s.d. ($n = 5$). (**h**) Average length of DRG neurite sprouting in the microfluidic chips. Mean ± s.d. ($n = 12$). Significant difference was from the nontreated control. *$P < 0.01$ compared with the nontreated control; Student's _t_-test.

The GNC–siRNA entered the Panc-1 cells in much larger quantity than free siRNA (Fig. 3b and Supplementary Fig. 10). In contrast, free siRNA hardly entered the Panc-1 cells (Fig. 3b and Supplementary Fig. 10) due to their high molecular weight and negative charges[19]. This indicated that GNC–siRNA complex facilitate the siRNA internalization compared to free siRNA.

To study the lysosomal escape of GNC–siRNA, we stained the Panc-1 cells with LysoTracker and monitored the cells using confocal fluorescence images (Fig. 3c). After 1 h incubation, red (GNC–Cy5-siRNA) and green (LysoTracker) fluorescence co-localized in the Panc-1 cells (Fig. 3c), indicating that the GNC–Cy5-siRNA was located in the lysosomes. After 4 h incubation, the red fluorescence (GNC–Cy5-siRNA) separated from the green fluorescence (LysoTracker; Fig. 3c), indicating that the GNC–Cy5-siRNA can escape from the lysosomes into the cytoplasm. We quantified the co-localization of GNC–siRNA with lysosomes over incubation time (Supplementary Fig. 11a); the result was consistent with the observation in Fig. 3c. And the escape ratio of GNC–siRNA from lysosomes increased from 40 to 76% from 3 to 6 h (Supplementary Fig. 11b). Since free siRNA

hardly entered the Panc-1 cells, it was hard to detect the co-localization of free siRNA with lysosomes (Supplementary Fig. 12). Thus, we believe that siRNA escaped from the lysosomes in the form of GNC–siRNA, and the GNC–siRNA was released into the cytoplasm for siRNA-mediated gene silencing.

**Gene knockdown of GNC–siRNA complex in Panc-1 cells.** We assessed the knockdown effect of the GNC–siRNA complex by measuring _NGF_ gene silencing in Panc-1 cells (Fig. 3d,e). The GNC–siRNA (100 nM siRNA equivalent) exhibited greater inhibition on _NGF_ mRNA expression, with 75% downregulation of _NGF_ mRNA expression compared to the nontreated control (Fig. 3d). Free siRNA (100 nM) showed slight inhibition, with 17% reduction of _NGF_ mRNA expression. In contrast, GNC–nonsense siRNA (GNC–nsRNA) complex did not significantly inhibit _NGF_ mRNA expression (Fig. 3d). Accordingly, the expression level of NGF protein in Panc-1 cells with GNC–siRNA was significantly lower than nontreated control and free siRNA (Fig. 3e). In addition, we also confirmed the gene

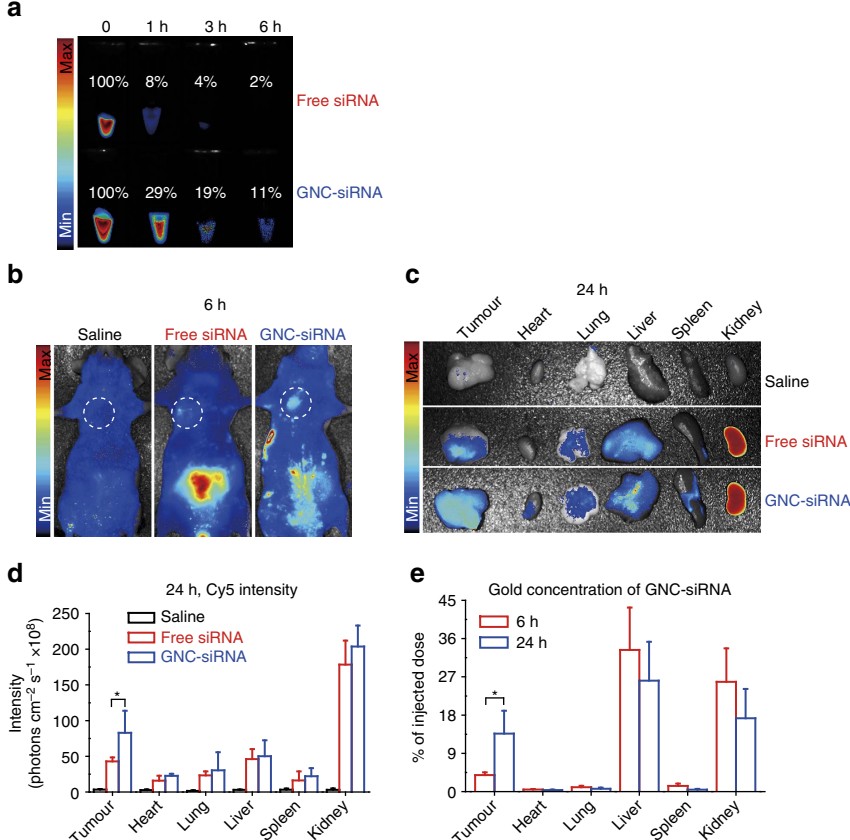

**Figure 5 | The circulation time and tumour targeting of GNC–siRNA complex in vivo.** (**a**) The circulation time of free siRNA and GNC–siRNA complex in blood. Cy5 dye-labelled siRNA (Cy5-siRNA) was used for visualization. Blood was drawn from mice after tail vein injection with various Cy5-siRNA formulations (30 μg Cy5-siRNA per mouse equivalent) at different time and imaged under a fluorescence imaging system. The fluorescence intensity of 0 h in each group was normalized as 100%. (**b**) *In vivo* tumour targeting. Balb/c nude mice with subcutaneous Panc-1 tumours were injected with different Cy5-siRNA formulations (30 μg Cy5-siRNA per mouse equivalent) via tail vein. After 6 h, fluorescence images of the mice were acquired with *in vivo* fluorescence imaging system, the white circles indicated the regions of subcutaneous tumours. (**c**) Tumour targeting by *ex vivo* imaging. After 24 h of Cy5-siRNA injection, major organs and tumours were isolated from mice for *ex vivo* fluorescence imaging. (**d**) The fluorescence intensity of Cy5-siRNA in major organs and tumours at 24 h after intravenous injection. *P < 0.01 compared with free siRNA. (**e**) The concentration of gold in the major organs and tumours (expressed as % of given dose) at 6 and 24 h post injection of GNC–siRNA complex by ICP-MS. Fluorescence images and ICP-MS analysis confirmed the accumulation of GNC–siRNA complex into the tumour sites. Mean ± s.d. (n = 4). *P < 0.01; Student's t-test.

knockdown efficiency of GNC–siRNA complex by sequence of siRNA-#5 (GNC–siRNA-#5; Supplementary Fig. 13). These results suggest that *NGF* siRNA can silence NGF expression in a highly sequence-specific manner. The GNC–siRNA protected the siRNA from degradation, facilitated the cellular uptake, and the GNC–siRNA can escape from the lysosomes to the cytoplasm for effective siRNA-mediated gene silencing.

**Cytotoxicity of GNC–siRNA complex in vitro.** We tested the cytotoxicity of the GNC–siRNA complex. In cell viability test, the GNC–siRNA complex was non-toxic to Panc-1 cells at tested concentrations from 0 to 5,000 nM of siRNA (Fig. 4a). In contrast, at the same concentration, the Lipofectamine 2000 transfection reagent-mediated siRNA delivery (Lipo2000-siRNA) induced significant toxicity to Panc-1 cells (Fig. 4a). This result was attributed to the good biocompatibility of gold and high loading ratio of siRNA by GNCs.

**Proliferation of Panc-1 cells.** We studied the effects of GNC–siRNA on the proliferation of Panc-1 cells. In proliferation assay, after 72 and 96 h in culture, Panc-1 cells treated with GNC–siRNA (100 nM siRNA) showed less proliferation than

nontreated cells and cells treated with GNC–nsRNA (Fig. 4b). This result indicates that *NGF* knockdown by GNC–siRNA complex inhibited the proliferation of Panc-1 cells.

**Migration of Panc-1 cells.** We evaluated the effect of *NGF* silencing on Panc-1 cell migration. We developed microfluidic chip assay to monitor cell migration (Fig. 4c)[35,36]. We evaluated the gap between the two patterns of Panc-1 cells on the chips (Fig. 4d). After 48 h, the gaps from Panc-1 cells treated with GNC–siRNA remained open, but almost closed for the nontreated and GNC–nsRNA-treated Panc-1 cells (Fig. 4d). The extent of gap closure from GNC–siRNA complex was lower than the nontreated control and GNC–nsRNA at 24 and 48 h, respectively (Fig. 4e). These results suggest that *NGF* depletion with GNC–siRNA inhibited the migration of Panc-1 cells.

**Neurite sprouting in response to *NGF* knockdown by GNC–siRNA.** To investigate the *NGF* knockdown on neurite growth, we developed the co-culture of dorsal root ganglion (DRG) neurons and Panc-1 cells on microfluidic chips (Supplementary Fig. 14a)[37–39]. The DRG neurons and Panc-1 cells were seeded into the chips and allowed to adhere before the removal of the

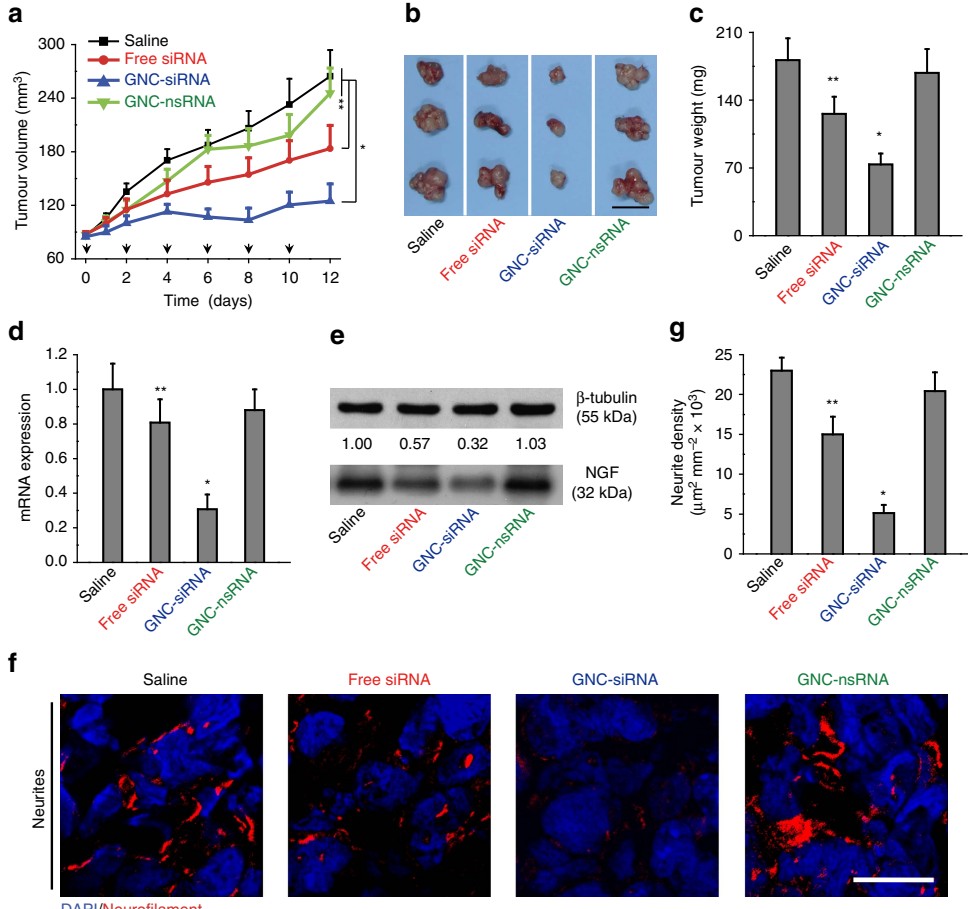

**Figure 6 | The antitumour and gene knockdown effects of GNC–siRNA complex in subcutaneous pancreatic tumours.** Panc-1 cells were injected into the flank of Balb/c nude mice to form subcutaneous tumours. When the tumours reached about 5 mm in diameter, the animals received peritumoral injections of various formulations every 2 days for six injections. (**a**) Tumour growth curve during the treatments. The black arrows indicated the days of injection. (**b**) *Ex vivo* tumour image and (**c**) tumour weight at the end of experiment. Scale bar, 1 cm. (**d**) Expression level of *NGF* mRNA and (**e**) NGF protein level in subcutaneous tumours. (**f**) IF staining of neurites in tumour tissues with various siRNA treatments. Neurites were stained with neurofilament antibody (red), the cell nuclei were stained with 4,6-diamidino-2-phenylindole (DAPI; blue). Scale bars, 20 μm. (**g**) Quantification of neurite density in the subcutaneous tumours by counting the neurite area positive to neurofilament antibody per unit area. Mean ± s.d. ($n = 6$ per group). Significant difference was from the saline control, $*P < 0.01$, $**0.01 < P < 0.05$; Student's *t*-test.

cover (Supplementary Fig. 14b). The neurite sprouting from neuronal compartment was assessed. Panc-1 cells pretreated with GNC–siRNA induced less DRG neurite sprouting compared to the nontreated controls (Fig. 4f), due to the *NGF* depletion in Panc-1 cells by GNC–siRNA complex. In contrast, the application of GNC–nsRNA to Panc-1 cells had no obvious change to DRG neurite sprouting (Fig. 4f). The quantification showed that the GNC–siRNA treatment potently inhibited the neurite density (Fig. 4g) and decreased the average length of neurite sprouting (Fig. 4h) in the microfluidic chips.

**Blood circulation time of siRNA *in vivo*.** To investigate the circulation time of siRNA in blood, we injected various Cy5-siRNA formulations into Balb/c nude mice via tail vein (30 μg siRNA per mouse equivalent). The fluorescent intensity in each group was normalized by considering 0 h intensity as 100%. In GNC–siRNA treatment, the fluorescence signal of Cy5-siRNA still remained at a high level (11%) in blood after 6 h compared to 0 h (Fig. 5a). In contrast, free Cy5-siRNA rapidly degraded and disappeared from the blood, and almost no fluorescent signals of Cy5-siRNA can be detected in blood after 3 h (Fig. 5a). In the GNC–siRNA complex, the GNC core structures had an ion cloud due to the spacer that carriers positive

charges, which is associated with a high density of siRNA shell (Fig. 1a–c). This structure prevented the nucleases from accessing the buried siRNA in the spacer, thus inhibiting the enzymatic degradation of siRNA during circulation. As a result, the GNC–siRNA complex resulted in the increased stability of siRNA and prolonged the circulation lifetime of siRNA in blood (Fig. 5a).

**Tumour targeting of GNC–siRNA in subcutaneous tumour.** To investigate the biodistribution of GNC–siRNA *in vivo*, we intravenously injected different Cy5-siRNA formulations into Balb/c nude mice bearing subcutaneous Panc-1 tumours. After 6 h injection, the fluorescence imaging of the mice showed the GNC–Cy5-siRNA significantly accumulated into the tumour region with a higher degree than the free siRNA and the saline control (Fig. 5b). After 24 h injection, the fluorescent imaging *ex vivo* showed that tumours treated with GNC–Cy5-siRNA complex exhibited higher level of fluorescent intensity at tumour sites than those treated with free Cy5-siRNA (Fig. 5c,d). Meanwhile, inductively coupled plasma mass spectrometry (ICP-MS) confirmed the accumulation of gold element in GNC–siRNA complex into the tumour sites (Fig. 5e). ICP-MS measurements showed that the GNC–siRNA was mainly

eliminated by the liver and the kidneys (Supplementary Fig. 15). It agrees with previous report that the liver can rapidly uptake, degrade, inactivate and eliminate the nanomaterials[40], while the kidney is an excretory organ where the nanoparticles could be excreted into the urine[41], (Supplementary Fig. 15, right panel). Both enhanced fluorescence intensity of Cy5-siRNA and gold at the tumour regions confirmed an increased accumulation of the GNC–siRNA into the tumour sites, presumably due to prolonged

circulation time of GNC–siRNA in blood circulation, and the leaky tumour vasculature via the enhanced permeation and retention effect[42].

**Antitumour efficiency**. We investigated the antitumour and gene knockdown effects of GNC–siRNA complex in three pancreatic tumour models, including subcutaneous model, orthotopic model and patient-derived xenograft (PDX) model.

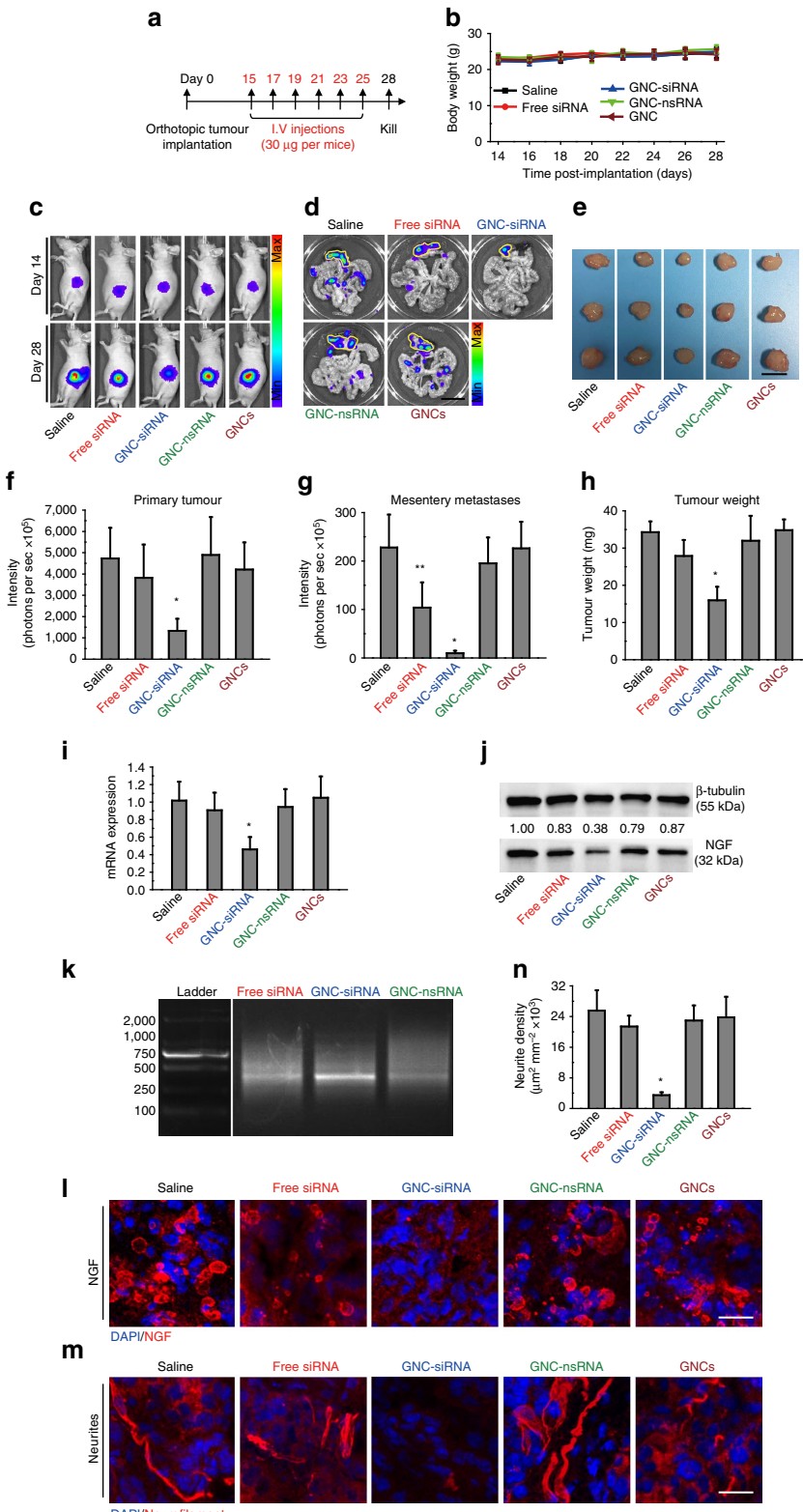

**Antitumour and gene knockdown effect in subcutaneous model.**
We injected the Panc-1 cells into right flank of Balb/c nude mice to form subcutaneous tumours, and examined the antitumour effects with peritumoral injections of various siRNA formulation (4 μg siRNA per mouse equivalent). The GNC–siRNA exhibited the most significant effects on tumour suppression, whereas the free siRNA injection inhibited the tumour growth to a limited degree (Fig. 6a). The tumour volumes for mice treated with GNC–siRNA and free siRNA were reduced by 52% and 30% compared to the saline control, respectively (Fig. 6a). In contrast, GNC–nsRNA treatment had no significant effect on tumour growth (Fig. 6a). The tumour sizes and tumour weight in GNC–siRNA group were smallest among all groups (Fig. 6b,c).

We evaluated the GNC–siRNA complex in depleting the expression level of *NGF* mRNA and NGF protein in tumours. The GNC–siRNA showed a greater inhibitory effect on *NGF* mRNA expression, with 69% reduction of *NGF* mRNA relative to the saline control (Fig. 6d). Free siRNA suppressed about 21% of *NGF* mRNA in the tumours compared to the saline control (Fig. 6d). The intratumoral level of NGF protein decreased in an siRNA formulation-dependent manner, where GNC–siRNA group induced the greatest inhibition on NGF protein expression (Fig. 6e). The enhanced *NGF* silencing of GNC–siRNA was due to the enhanced cellular uptake of the GNC–siRNA complex than free siRNA, and the RNA interference (RNAi)-mediated gene silencing[18].

We used IF staining of neurofilament to explore the intratumoral neurogenesis (Fig. 6f,g). GNC–siRNA group showed a significant reduction of the density of intratumoral neurites, whereas free siRNA treatment reduced the intratumoral neurogenesis to a limited degree (Fig. 6f,g). The data in subcutaneous model reconfirmed our hypothesis that *NGF* silencing inhibits the progression of subcutaneous pancreatic tumours.

**Antitumour and gene knockdown effect in orthotopic model.**
We created orthotopic pancreatic tumour models to mimic tumour microenvironment that most resembled the *in vivo* scenario, and evaluated the antitumour effect of siRNA by systemic administration (30 μg siRNA per mice equivalent) for six times (Fig. 7a). No significant change of mouse body weights was observed for the mice during the experimental period (Fig. 7b), indicating that there was no apparent systemic toxicity caused by different siRNA treatments.

We acquired the *in vivo* bioluminescence imaging (BLI) of mice to evaluate the tumour progression (Fig. 7c). GNC–siRNA group remarkably inhibited tumour growth in Balb/c nude mice (Fig. 7c,f). Different from the case of peritumoral injection of siRNA into subcutaneous tumour model, free siRNA did not

lead to a significant decrease in the orthotopic tumour growth compared to the saline control (Fig. 7c,f), most likely due to the short circulation time and the rapid degradation of the free siRNA in the blood stream. Tumours in saline control, GNC–nsRNA and GNC groups grew steadily from week 2 (Fig. 7c,f). To assess the metastases of pancreatic tumours, we excised the tumours in pancreas with associated mesenteries for *ex vivo* BLI images (Fig. 7d). Compared to the saline group, the GNC–siRNA group markedly reduced the mesentery metastases, and almost no detectable metastasis was observed by BLI imaging (Fig. 7d,g). At day 28, the tumour sizes and tumour weight treated by GNC–siRNA group were the smallest among all groups (Fig. 7e,h). The GNC–nsRNA did not show a significant inhibitory effect on tumour growth (Fig. 7e,h).

We examined the *NGF* knockdown in orthotopic tumours with siRNA treatments. GNC–siRNA group significantly reduced the *NGF* mRNA expression in orthotopic tumours compared to the saline control (Fig. 7i). Free siRNA did not induce a significant decrease in the *NGF* mRNA expression (Fig. 7i). The GNC–siRNA group efficiently inhibited the expression of NGF protein in orthotopic tumours, but free siRNA did not significantly reduce the NGF protein expression in tumours (Fig. 7j).

We used 5′-RACE assay to verify the *in vivo* RNAi mechanism of the GNC–siRNA complex. In GNC–siRNA-treated tumours, 5′-RACE analysis using human *NGF* gene-specific primers identified a PCR product of expected cleavage product with a molecular weight ∼370 bp (Fig. 7k). This band was not obvious in the free siRNA and in GNC–nsRNA-treated tumours (Fig. 7k). The cleavage products confirmed a sequence-specific, RNA-induced silencing complex-mediated activity in GNC–siRNA-treated tumours.

We confirmed the distribution of NGF in orthotopic tumours by IF staining (Fig. 7l). NGF expression was markedly suppressed by GNC–siRNA group compared to the strong expression of NGF protein in the saline control (Fig. 7l). The free siRNA did not induce a significant decrease of NGF protein expression in tumour tissues. In other groups, the expression of NGF was similar to that in the saline group. These results showed that GNC–siRNA complex can effectively decrease the NGF expression in tumour tissue.

We performed neurofilament staining to evaluate the effect of *NGF* knockdown on intratumoral neurite growth (Fig. 7m). The GNC–siRNA treatment demonstrated a remarkable inhibition of the intratumoral neurite growth (Fig. 7m). The quantification showed that the GNC–siRNA group reduced 85.7% neurite formation in the tumours compared to the saline control (Fig. 7n). In addition, we conducted a systematical study with various GNC–siRNA formulations in the orthotopic

**Figure 7 | The antitumour and gene knockdown effects of GNC–siRNA complex in orthotopic tumours.** (**a**) Scheme of siRNA treatment. Panc-1-luc cells were injected into the pancreas head of Balb/c nude mice to form orthotopic tumours. After 2 weeks, mice were divided into different groups. Mice received various formulations via tail veil injections for six times, and killed on day 28. (**b**) The changes of the mouse body weight during treatments. (**c**) *In vivo* whole-body bioluminescence images of mice on day 14 and day 28, which indicated the tumour size before and after siRNA treatment. Bioluminescence signal was a result from the interaction of luciferase from Panc-1-luc cells with D-luciferin injected into the mice before imaging. (**d**) *Ex vivo* bioluminescence images of orthotopic pancreatic tumours and tumour metastases into mesenteries on day 28. Yellow lines enclosed the locations of primary tumours in the pancreas. Scale bar, 1 cm. (**e**) Tumour images on day 28. Scale bar, 5 mm. (**f**) Quantification of *in vivo* bioluminescence to evaluate the primary tumours in mice on day 28. (**g**) Quantification of tumour metastases by the sum of *ex vivo* bioluminescence detected in the mesenteries on day 28. (**h**) Weight of the isolated tumours. (**i**) *NGF* mRNA and (**j**) NGF protein expression levels in orthotopic tumours. (**k**) Confirmation of RNAi-mediated mechanism of action with GNC–siRNA by 5′-RACE assay. A 2% agarose gel electrophotosis showed ∼370 bp RNA-induced silencing complex-mediated cleavage product for *NGF* siRNA in pancreatic tumours. Only tumours treated with GNC–siRNA complex showed the cleavage product at 370 bp. The left column was the DNA ladder with described molecular weights. (**l**) IF images of NGF protein (red) in the orthotopic tumours. (**m**) IF images of neurites (red) in the orthotopic tumours. The cell nuclei were stained with 4,6-diamidino-2-phenylindole (DAPI; blue). Scale bars, 20 μm. (**n**) Quantification of the neurite density in the orthotopic tumours. Mean ± s.d. ($n = 6–9$ per group). Significant difference was from the saline control, *$P < 0.01$, **$0.01 < P < 0.05$; Student's *t*-test.

tumour models. GNC–siRNA-#2 and GNC–siRNA-#5 complexes showed more effective antitumour effects and *NGF* gene knockdown effects than GNC–siRNA-#3 in the orthotopic tumours (Supplementary Fig. 16).

**Antitumour and gene knockdown effect in PDX model.** We conducted PDX tumour models in Balb/c nude mice to mimic naturally occurred tumours in human patients[43,44]. The procedure was illustrated in Fig. 8a. The mice received systemic

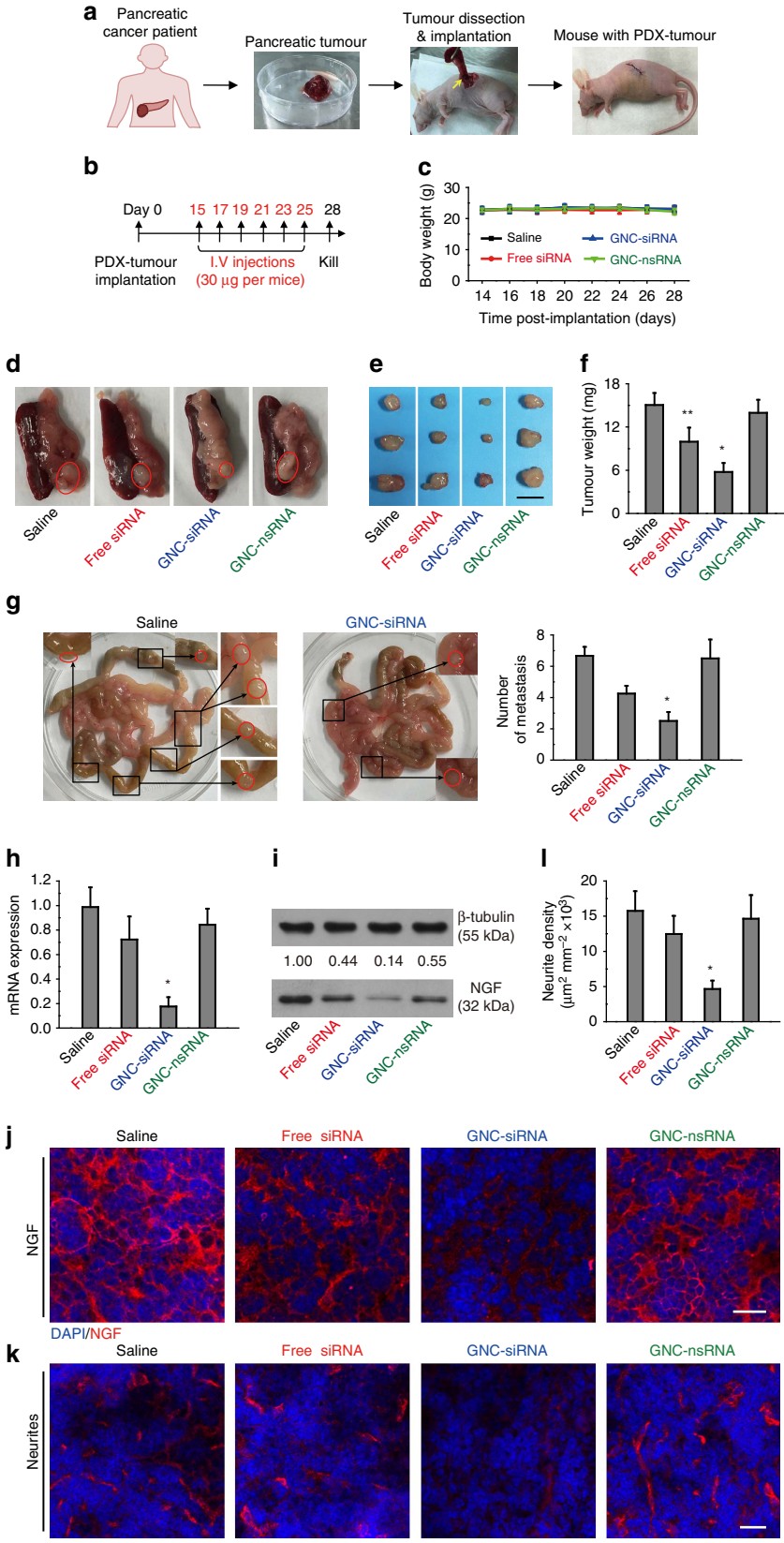

administration of various siRNA formulations (30 μg siRNA per mouse equivalent) for six times (Fig. 8b). No significant changes of mouse body weight was observed for the mice during the period of treatment (Fig. 8c), indicating no apparent systemic toxicity for all treatment groups. At the end of the experiments, GNC–siRNA group resulted in a more significant decrease in tumour sizes compared to the saline control (Fig. 8d), whereas free siRNA inhibited tumour growth to a limited degree (Fig. 8d); GNC–nsRNA treatment had no significant antitumour effect (Fig. 8d). The tumour image and weight showed that the GNC–siRNA-treated tumours were significantly smaller than those from the saline control, free siRNA and GNC–nsRNA group (Fig. 8e,f). We analysed the tumour metastases into mesenteries (Fig. 8g), the GNC–siRNA group significantly reduced the mesentery metastases compared to the other groups (Fig. 8g).

We assessed the NGF expression in the PDX tumour tissues. The GNC–siRNA group induced the most significant inhibition on the expression level of both *NGF* mRNA and NGF protein in the PDX tumours (Fig. 8h,i). We confirmed the distribution of NGF protein in orthotopic pancreatic tumours by IF staining (Fig. 8j), where the NGF level was markedly suppressed by GNC–siRNA group compared to the saline control (Fig. 8j). IF staining of neurofilament confirmed that the intratumoral neurite density was correlated with the intratumoral NGF protein expression (Fig. 8k,l), where the GNC–siRNA group remarkably reduced the intratumoral neurite growth (Fig. 8k,l).

***In vivo* toxicity of GNC–siRNA.** Free siRNA duplexes are potent to activate the mammalian innate immune system, and lead to systemic inflammation *in vivo* through inducing high levels of inflammatory cytokines[45,46], thus the inflammatory responses by siRNA can result in significant toxicities *in vivo*. We used the GNC–siRNA complex to reduce the free siRNA-induced immunostimulation. We investigated the *in vivo* toxicity of GNC–siRNA complex after multiple dosing treatment. From the haematoxylin and eosin-stained sections of heart, liver, spleen, lung and kidney, no obvious histological changes were observed between the GNC–siRNA group and the control groups (Supplementary Fig. 17a). However, many tumour cells were dead in the GNC–siRNA group (Supplementary Fig. 17a, right panel). The staining of apoptosis and necrosis markers confirmed that the GNC–siRNA-treated tumours induced a higher apoptosis and necrosis level than other groups (Supplementary Methods and Supplementary Fig. 17b–d). These results indicated the antitumour efficacy with minimal organ toxicity from the GNC–siRNA group. The GNC–siRNA group kept the parameters of liver and kidney functions within the normal range (Supplementary Table 6). However, free siRNA induced a relatively high level of alanine amino transferase (ALT) and aspartate amino transferase (AST; Supplementary Table 6), indicating hepatic dysfunction from the free siRNA treatment[47].

In addition, at the therapeutic doses, no significant immune response was induced by GNC–siRNA group, as the production of serum cytokines, including interleukin-6 (IL-6) and tumour necrosis factor-α (TNF-α) was similar to the saline control group (Supplementary Fig. 18). However, the free siRNA group induced a relatively high immune responses by production of more IL-6 and TNF-α (Supplementary Fig. 18)[45]. Together, GNC–siRNA complex showed high safety and low toxicity *in vivo* compared to free siRNA.

## Discussion

In this study, we have developed an siRNA-based gene regulator, GNC–siRNA complex, to effectively inhibit pancreatic cancer progression by depleting *NGF*. The GNC–siRNA complex successfully downregulated the *NGF* gene via siRNA/RNA-induced silencing complex pathway (Fig. 9)[18]. And we confirmed the *in vivo* RNAi-mediated mechanism from the GNC–siRNA complex by 5′-RACE assay (Fig. 7k).

GNCs with sizes smaller than 3 nm are relatively new as a class of gold nanomaterials, which show great promises in biological applications[48–50]. In this study, cationic GNCs were developed

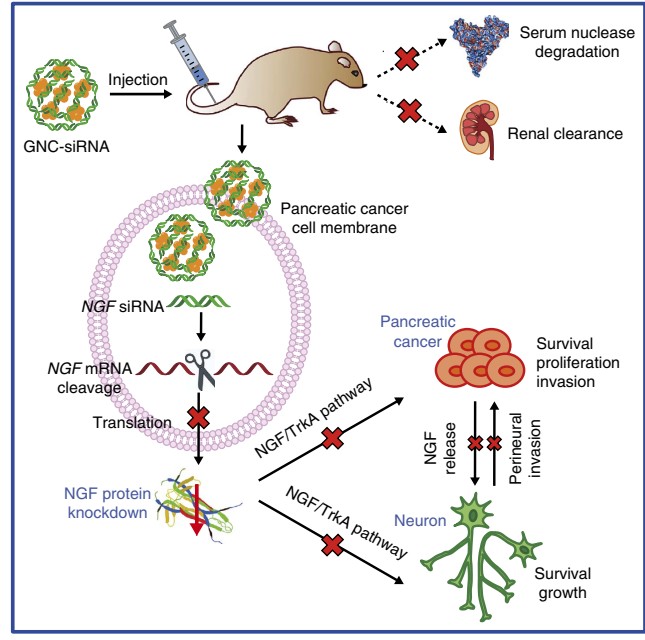

**Figure 9 | Delivery mechanism of GNC–siRNA complex for *NGF* silencing and pancreatic cancer therapy.** The GNC–siRNA complex protects the *NGF* siRNA from serum nuclease degradation and renal clearance, enhances the accumulation of *NGF* siRNA to tumour cells, which allows efficient *NGF* silencing in pancreatic tumours. The downregulated NGF expression inhibits the pancreatic cancer progression and neurogenesis in pancreatic tumour microenvironment.

**Figure 8 | The antitumour and gene knockdown effects of GNC–siRNA complex in orthotopic pancreatic PDX tumours.** (**a**) Scheme of the establishment of PDX tumour model. Patient-derived pancreatic tumours were trimmed, cut into fragments with similar sizes and transplanted into the pancreas head of the Balb/c nude mice. (**b**) Scheme of siRNA treatments. Two weeks after PDX tumour transplantation, mice were randomly divided into different groups and injected with various siRNA formulations through tail veil for six injections. The mice were killed on day 28. (**c**) The effect of different siRNA treatments on the changes of mouse body weight. (**d**) Representative images of the orthotopic PDX tumours in pancreas with associated spleen on day 28, tumours were indicated in red circles. (**e**) Tumour images and (**f**) tumour weight measured on day 28. Scale bar, 5 mm. (**g**) Representative images and quantification of tumour metastases into mesenteries. Tumour metastases were magnified and indicated by red circles. (**h**) *NGF* mRNA and (**i**) NGF protein level in the PDX tumours. (**j**) Representative images of the IF staining of NGF protein (red) in the PDX tumours. (**k**) Images of IF staining of neurites in PDX tumours. Neurites were stained with neurofilament antibody (red). The cell nuclei were stained with 4,6-diamidino-2-phenylindole (DAPI; blue). Scale bars, 20 μm. (**l**) Quantification of neurite density in the PDX tumours. Mean ± s.d. (*n* = 5–6 per group). Significant difference was from the saline control, *$P < 0.01$, **$0.01 < P < 0.05$; Student's *t*-test.

with simply one-step reaction; moreover, the GNC–siRNA complexes do not require additional chemical modifications to facilitate entry into cells. The use of GNCs in delivering siRNA is advantageous: they increase the stability of the siRNA in serum, prolonged the circulation time of the siRNA in blood and enhance the cellular uptake and tumour accumulation of the siRNA. The most striking feature of the GNCs in this study is their extremely high siRNA-loading capacity; the GNCs can load as high as 226 µmol siRNA per g GNCs (Supplementary Table 3). To our knowledge, the GNCs have the highest loading of siRNA compared to other materials reported to date. Previous reports developed 13 nm-diameter GNPs for delivery of siRNA in several disease models[31,32]; the GNPs had a loading of 40 siRNA strands per particle (equal to 3 µmol siRNA per g GNPs)[32]. Thus, the GNC–siRNA complex here provided much higher loading capacity of siRNA, most likely due to the smaller size, the larger specific surface area and the positive charge on the surface of GNCs (Fig. 1).

Pancreatic cancer is one of the deadliest human cancers, whose progression is highly dependent on nervous microenvironment. In the pancreatic tumour microenvironment, NGF binds tyrosine kinase A (TrkA) receptors, leading to the activation of various downstream signalling cascades, such as PI3K (phosphatidylinositol 3-kinase)–Akt signalling pathway, which contributes to survival of both cancer and neuronal cells[51], another one is RAS–MAPK (mitogen-activated protein kinase signalling pathway, which leads to the survival, proliferation and invasion of cancer cells[51]. As a result, the NGF depletion in Panc-1 cells with GNC–siRNA complex significantly reduced the growth, proliferation and migration of pancreatic cancer cells, as well as reduced the neurite sprouting in vitro (Fig. 4). In the in vivo models, GNC–siRNA complex enhanced the blood circulation and the accumulation of NGF siRNA to tumour sites (Fig. 5), induced significant NGF silencing (Figs 6–8). The NGF depletion with GNC–siRNA complex inhibited both pancreatic tumour growth and intratumoral neurite growth (Fig. 9); in turn, the reduced neurogenesis by GNC–siRNA complex reduced the metastasis of pancreatic cancers via nerves by the process of perineural invasion (Fig. 9). The pancreatic tumour progression was correlated well with the intratumoral neurogenesis (Figs 6–8). The density of neurites in the tumours was significantly decreased with GNC–siRNA complex where the tumour growth was significant inhibited (Figs 6–8).

Despite the effectiveness in subcutaneous tumour model with intratumoral injection (Fig. 6), the failure of free siRNA to improve antitumour effect in orthotopic model (Fig. 7) suggested that the GNC–siRNA complex can resist serum nucleases degradation of naked siRNA, prolong the circulation lifetime of siRNA in blood, enhance the accumulation of siRNA to tumours and knockdown the targeted NGF gene, thus inhibit the orthotopic pancreatic tumour progression. These data also pointed out the importance of testing the efficacy of treatment on multiple types of tumour models.

Our results indicate that the GNC–siRNA complex induced efficient NGF depletion without apparent toxicity both in vitro (Fig. 4a) and in vivo (Supplementary Figs 17 and 18, and Supplementary Table 6), suggesting the feasibility of using GNCs as carriers for NGF siRNA delivery and anti-neurogenic cancer therapy. However, the detailed pathway of the entry of GNC–siRNA complex into cells should be investigated in our future works. Moreover, the GNC–siRNA complex passively targeted and accumulated in tumour regions via enhanced permeation and retention effect in this study. Future work will decorate the GNC–siRNA complex with active tumour-targeting molecules to enhance the active tumour targeting. We will also consider modifying the GNC–siRNA complex with hydrophilic

polymers such as poly(ethylene glycol) to further prolong their blood circulation time. The genetically engineered mouse models of pancreatic cancer, such as the LSL-Kras[G12D/+]; LSL-Trp53[R172H/+]; Pdx-1-Cre (KPC) mice[52,53], can generate the dense desmoplasia, which defines pancreatic cancer and affects its most important pharmacological features[52,54]. In the future study, we will use the genetically engineered mouse models of pancreatic cancer to explore the translatability of the GNC–siRNA complex.

To conclude, we have developed biocompatible GNCs as a delivery platform for siRNA (GNC–siRNA complex) to target the tumour–neuron interaction for pancreatic cancer treatment via NGF depletion. The GNC–siRNA complex provides the NGF siRNA not only a protective structure against the harsh biological environment but also an opportunity for passive targeting to the tumour tissues. The GNC–siRNA complex enhances the specific NGF gene knockdown in Panc-1 cells in vitro and in pancreatic tumours in vivo, leading to effective suppression of tumour growth in three pancreatic tumour models without adverse effects and toxicity. Together, GNC-assisted delivery of NGF siRNA is a promising therapeutic approach for pancreatic cancer treatment by targeting the interactions between the tumours and the nervous microenvironment.

## Methods

**Materials.** All chemicals were commercially available and used without further purification. Hydrogen tetrachloroaurate trihydrate (HAuCl$_4$.3H$_2$O) and GSH were obtained from Sigma-Aldrich. Custom-designed oligoarginine peptide CRRRRRRRRR (CR$_9$) was supplied by GL Biochem Ltd. (Shanghai, China). Ultrapure water (Milli-Q) with a resistivity of 18.2 MΩ was used as the general solvent throughout the study.

**Preparation of GNCs.** We synthesized the fluorescent GNCs from one-step reduction of Au$^{3+}$ in the presence of thiolate containing GSH and oligoarginine as previously reported[33]. In brief, HAuCl$_4$ solution (100 mM), GSH (150 mM) and oligoarginine CRRRRRRRRR (CR$_9$, 75 mM) were mixed with ultrapure water at 25 °C. The mixture was heated to 70 °C under gentle stirring (500 r.p.m.) for 24 h. An aqueous solution of GNCs with light-green colour was formed. The GNC solution could be stored at 4 °C for 6 months with negligible changes in their optical properties.

**Preparation of GNC–siRNA complex.** The positively charged GNCs (1 µg ml$^{-1}$) were mixed with siRNA solution in ultrapure water, and shaken on a bench-top shaker for 1 h to complete the binding of siRNA onto the GNCs via electrostatic interaction. The NGF siRNA was added into the GNC solution in different concentrations to determine the saturated concentration of siRNA solution, with the weight ratio of siRNA to GNCs varied from 0:1 to 100:1. The prepared samples were abbreviated as GNC–siRNA. The GNCs binding with nsRNA were abbreviated as GNC–nsRNA and served as control siRNA.

**Characterization of GNC–siRNA conjugates.** The size distribution, polydispersity index and surface charge of the prepared nanomaterials were determined using dynamic light scattering (NanoZS, Malvern). For morphology characterization, the samples were examined by transmission electron microscope (TEM, Tecnai G2 20S-TWIN) and cryogenic TEM (CryoTEM, Talos F200C).

XPS was performed to detect elemental composition of the samples. The sample solution was dropped onto an aluminium foil and freeze-dried. The analyses were performed on a Thermo Scientific ESCALAB 250XI photoelectron spectrometer with a monoachromatized X-ray source (powered at 20 mA and 10 kV). Data treatment was performed with the CasaXPS programme.

Electrophoretic mobility shift assay with polyacrylamide gel electrophoresis was used to evaluate the protection of siRNA against serum nuclease degradation. Free siRNA and GNC–siRNA complex (100 nM siRNA equivalent) were incubated at 37 °C in DMEM containing 10% fetal bovine serum (FBS) for 0 min, 15 min, 30 min, 45 min, 1 h and 6 h. A volume of 10 µl of each sample was loaded into each well of a 20% (w/v) polyacrylamide gel, and the voltage was applied at 160 V for 45 min. The resulting gel was stained with GelRed (Sigma) and visualized on a Gel Doc imaging system (Bio-Rad). Uncropped gel was shown in Supplementary Fig. 19.

**In vitro studies.** *Cell culture*. Human pancreatic Panc-1 cancer cells were obtained from the Institute of Basic Medical Sciences, Chinese Academy of Medical Sciences. Cells were maintained in DMEM medium supplemented with 10% FBS and 1% penicillin–streptomycin (PS). Panc-1 cells stably transfected with luciferase gene

(Panc-1-luc cells) were obtained from the Institute of Basic Medical Sciences, Chinese Academy of Medical Science, and maintained in DMEM with 10% FBS, 1% PS and 600 μg ml$^{-1}$ G418. The cell lines were tested to be mycoplasma-free cells, and routinely maintained in a humidified atmosphere containing 5% CO$_2$ at 37 °C.

*Screening of siRNA sequences.* We evaluated the *in vitro* transfection efficiency of *NGF* siRNA sequences by measuring *NGF* mRNA level in Panc-1 cells. Five siRNA duplexes against human *NGF* and a nsRNA as controls (Supplementary Table 1) were synthesized by Ribobio (Guangzhou, China). For transfection, Panc-1 cells were plated at a density of $3 \times 10^5$ cells per well in a six-well plate. When the cells were 50% confluent, siRNA duplex (100 nM for each) was mixed with 5 μl Lipofectamine 2000 Reagent (Invitrogen) and incubated with Panc-1 cells for 48 h according to the manufacturer's protocol. Each treatment was conducted at least in triplicate. The expression levels of *NGF* mRNA in Panc-1 cells were analysed by quantitative real-time PCR (RT–PCR). The most effective siRNA sequence that result in the greatest knockdown of *NGF* mRNA was selected for further study (*NGF* siRNA-#2).

*Screening of nanomaterials.* We prepared different nanomaterials conjugated with *NGF* siRNA as described in Supplementary Methods. The nanomaterials–*NGF* siRNA conjugates were screened for their knockdown efficiency of *NGF* mRNA. Each nanomaterials–*NGF* siRNA conjugate was incubated with Panc-1 cells (100 nM siRNA for each) in complete medium for 24 h, after which the cells were rinsed with PBS and allowed to incubate for another 24 h in fresh complete medium. At least three independent trials were carried out for each nanomaterials–siRNA conjugate being tested. *NGF* mRNA expression was quantified in triplicate using RT–PCR. Of the nanomaterials screened, GNC–siRNA complex resulted in the greatest downregulation of *NGF* expression and was used for all future studies.

*Cellular uptake and lysosomal escape of siRNA.* For better observation, *NGF* siRNA was labelled with a far-red fluorescent dye Cy5 (excitation/emission at 649/670 nm) at the 5′-end of the sense strand (Cy5-siRNA) (Ribobio, Guangzhou). For cellular uptake, the Panc-1 cells were seeded at a density of $1 \times 10^5$ cells per well in a 20 mm confocal dish (Nunc) for 24 h. The cells were incubated with free Cy5-siRNA and GNC–Cy5-siRNA (100 nM Cy5-siRNA equivalent) in culture medium at 37 °C for 1 h. The cells were washed with PBS, fixed in 4% paraformaldehyde (PFA) solution for 15 min, counterstained with 4,6-diamidino-2-phenylindole and imaged with microscopy using a 633 nm laser excitation (Carl Zeiss LSM710).

To study the siRNA escape from lysosomes, the Panc-1 cells were seeded at a density of $1 \times 10^5$ cells per well into confocal dish for 24 h. The Panc-1 cells were stained with LysoTracker Green (Invitrogen, 75 nM) for 1 h. Then the Panc-1 cells were incubated with GNC–Cy5-siRNA and free Cy5-siRNA (100 nM Cy5-siRNA equivalent) at 37 °C for 1 h. The cells were rinsed and incubated in fresh medium for another 1–6 h. At different time points, the cells were fixed by 4% PFA and observed by confocal microscopy. The excitation wavelengths of LysoTracker Green and Cy5-siRNA were set as 488 and 633 nm, respectively.

*Real time–PCR.* Total RNA of the cells was extracted using TRIzol reagent (Invitrogen), the RNA was converted to cDNA and the RT–PCR was performed using *NGF* gene-specific primers (Supplementary Table 2) and the 7900 Fast RT–PCR system (Applied Biosystems) to determine *NGF* mRNA expression. The quantification was normalized to the endogenous control gene *RPLP0* (acidic ribosomal protein P0).

*Western blot.* Total proteins were extracted from transfected cells. A unit of 50 μg proteins for each treatment was separated on 10% SDS gel and transferred to a polyvinylidene difluoride membrane. The membranes were probed with rabbit monoclonal antibody against NGF (Abcam, Ab52918, 1:1,000) at 4 °C overnight, followed by incubation with horseradish peroxidase-linked goat anti-rabbit secondary antibody (Invitrogen, G-21234, 1:2,000) for 1 h at 37 °C. Beta-tubulin was used as an internal control (Invitrogen, MA5-16308, 1:5,000). The western blot signals were detected using a ChemiDoc XRS System (Bio-Rad). The band intensity of proteins was quantified using ImageJ software. Uncropped blots were presented in Supplementary Fig. 19.

*In vitro cytotoxicity.* For the *in vitro* cytotoxicity test, Panc-1 cells were plated in 96-well plates at a density of $3 \times 10^3$ cells per well. When the cells were 50% confluent, the cells were treated with GNC–siRNA or Lipofectamine 2000-transfected siRNA, with siRNA concentration ranging between 0 and 5,000 nM for 24 h. The proportion of viable cells was evaluated using a CCK-8 assay according to the manufacturer's instructions (Dojindo, Japan).

*In vitro proliferation assay.* Panc-1 cells were plated in 96-well plates ($3 \times 10^3$ cells per well) in complete medium. When the cells were 50% confluent, the cells were incubated with GNC–siRNA or GNC–nsRNA (100 nM siRNA equivalent). After 24 h, the cells were rinsed with PBS and incubated in fresh complete medium. Then the proliferation of Panc-1 cells was assessed during 72 h using a CCK-8 assay according to the manufacturer's instructions.

*In vitro migration assay.* We used microfluidic chip-based cell patterning to study the Panc-1 cell migration[35,36]. The chips contained two parallel channels with 450 μm gap (Fig. 4c). Panc-1 cells were pretreated with different siRNA formulations (100 nM siRNA equivalent for each) in complete medium for 48 h. The cells were rinsed and incubated with mytomycin C (5 μg ml$^{-1}$) for 1 h to inhibit the cancer cell proliferation. The treated Panc-1 cells at a density of $6 \times 10^6$ cells per ml were introduced into microfluidic channels and allowed to adhere for 6 h. Then polydimethylsiloxane cover was peeled off to form cell patterns on the surface. The cells were rinsed with medium, and the migration of Panc-1 cells was serially imaged on microscopy (Leica DMI6000 B) at different time points.

*In vitro co-culture of DRG neurons and Panc-1 cells.* We conducted the co-culture of DRG neurons and Panc-1 cells in microfluidic chips to study the sprouting of DRG neuron[37]. In brief, DRG neurons were isolated from the Sprague Dawley rats according to previously established techniques[39,55]. Panc-1 cells were pretreated with different siRNA formulations (100 nM siRNA equivalent) for 48 h. In co-culture assay, the DRG neurons ($6 \times 10^7$ cells per ml) and the Panc-1 cells ($6 \times 10^6$ cells per ml) were introduced into the left and the right channels of the chips, respectively (Supplementary Fig. 14a). The cells were allowed to adhere for 6 h before the removal of polydimethylsiloxane covers (Supplementary Fig. 14b). The cells were rinsed and the co-culture was maintained in Neurobasal medium containing 2% B27 supplement and 1% PS, for 3 days. The neural sprouting from DRG channels was labelled with anti-Tuj 1 antibody (Abcam, Ab18207, 1:200) and observed by microscope.

**In vivo studies.** *Animals.* Balb/c nude mice (8 weeks, male) were obtained from Vital River Laboratory Animal Center (Beijing, China) and raised in a specific pathogen-free environment. All animal studies were approved by the Institutional Animal Care and Use Committee, Institute of Process Engineering, Chinese Academy of Sciences (IACUC, IPE-CAS, IRB Number: SYXK 2014-0002). The animal studies were not blinded. Sample size was determined by power analysis. All animals were included in the analysis.

*Blood circulation time.* We labelled the *NGF* siRNA with Cy5 dye (Cy5-siRNA) to minimize the interference from autofluorescence in the *in vivo* imaging studies. To evaluate the circulation time, Balb/c nude mice were injected with free Cy5-siRNA and GNC–Cy5-siRNA (30 μg Cy5-siRNA per mouse equivalent) through tail vein. At 0, 1, 2, 3 and 6 h post injection, about 150 μl blood from the tail vein of mice was collected in heparin-treated tubes (BD), and 100 μl blood from each sample was transferred into plastic PCR tube (Axygen) and imaged with Maestro 2 fluorescence imaging system (CRi, USA), with an excitation:emission of 649:670 nm.

*Biodistribution of GNC–siRNA complex.* To study the tumour-targeting ability of GNC–siRNA complex, Balb/c nude mice with subcutaneous Panc-1 tumours were injected with saline, free Cy5-siRNA and GNC–Cy5-siRNA (30 μg Cy5-siRNA per mouse equivalent) through tail vein. At 1, 6 and 24 h post injection, the fluorescent images of the mice were imaged with Maestro 2 *in vivo* imaging system with a yellow filter (acquisition from 650 to 800 nm stepped in 10 nm increments). After 24 h, mice were killed, the major organs and tumours were dissected, and the *ex vivo* fluorescent images were acquired using the same system.

To further determine the biodistribution of gold in the GNC–siRNA complex after systemic injection, major organs, tumours, blood and urine were collected at 1, 6, 12, 24 and 48 h post injection. Samples were dried at 80 °C overnight, weighed and dissolved in 0.5 ml concentrated HNO$_3$ for 6 h. Then 0.5 ml concentrated HCl was added and the mixture was digested with a maximum temperature of 125 °C overnight. Afterwards H$_2$O was added into the vial to a total volume of 10 ml. All measurements of gold standards solution and samples were carried out with ICP-MS system using Rhenium as internal standard (20 parts per billion, ppb) (ICAP Q, Thermo). The ICP-MS was conducted in triplicate. The gold concentration was expressed as percentage of the injected dose for the organs and tumours, and expressed as nanogram per gram for blood and urine.

*Antitumour effects in vivo.* We conducted three pancreatic tumour models, including subcutaneous tumour model, orthotopic tumour model and PDX tumour model, to investigate the *NGF* gene silencing and antitumour effects of the GNC–siRNA complex.

*Subcutaneous tumour model.* We injected $1 \times 10^6$ Panc-1 cells into the flank of Balb/c nude mice to form subcutaneous tumours. When the tumour sizes reached about 5 mm in diameter, mice were sorted to give nearly identical mean tumour sizes, and different siRNA formulations (4 μg siRNA per mouse equivalent) were injected as close as possible to the tumours ($n = 6$ mice per group). Saline was injected as the control group. The injections were repeated every 2 days for six injections. The tumour volumes were determined by measuring the length ($l$) and width ($w$) and calculating the volume as $V = l.w^2/2$. On killing of animals, the tumour xenografts were excised, weighed and imaged.

*Orthotopic tumour model.* We created orthotopic tumours in Balb/c nude mice to mimic the natural pancreatic cancer microenvironment. Briefly, mice were anaesthetized, $1 \times 10^6$ Panc-1-luc cells were injected into the pancreas head of Balb/c nude mice using a micro-syringe (Hamilton). Two weeks after the tumour implantation, the mice were randomly divided into different groups: saline; free siRNA; GNC–siRNA; GNC–nsRNA; and GNCs ($n = 6$–9 mice per group), and treated with systemic administration of different siRNA formulations (30 μg siRNA per mice equivalent) via tail vein injection every 2 days for six injections. The body weights of the mice were monitored over the experiment period.

The growth of orthotopic tumours was monitored by BLI (Xenogen IVIS Spectrum). Bioluminescent signal was a result due to the reaction of luciferase from Panc-1-luc cells with the D-luciferin (150 mg kg$^{-1}$, Promega), which was intraperitoneally injected into mice 8–10 min before the imaging. BLI images of the mice were performed at day 14, 21 and 28 post-tumour implantation. On day 28, the mice were killed and tumour xenografts with associated mesenteries were collected for *ex vivo* BLI analyses. For *ex vivo* imaging, organs of interest were

immersed in saline solution containing 300 µg ml$^{-1}$ D-luciferin, and imaged for 1 to 2 min.

*PDX tumour model.* We conducted PDX tumour model in Balb/c nude mice to authentically mimic the pancreatic tumour growth in patients[56,57]. Human pancreatic tumours were obtained in the Chinese PLA General Hospital, with approval of the Institutional Review Board of Chinese PLA General Hospital and with the patients' informed consent ($n = 3$). For each patient's tumour, we did a series of experiments to compare the effects of GNC–siRNA complex. For each set of experiment, the surgically resected pancreatic tumour (about 1 cm$^3$) was trimmed, cut into approximate $3 \times 3 \times 3$ mm$^3$ fragments and transplanted into the pancreas head of Balb/c nude mice. The implantation process was finished within 3 h after surgery. Two weeks after the tumour transplantation, the mice were randomized into different groups: saline; free siRNA; GNC–siRNA; and GNC–nsRNA ($n = 5$–6 mice per group), and treated with different siRNA formulations (30 µg siRNA per mouse equivalent) via tail vein injection every 2 days for six injections. On day 28, the mice were killed, tumour xenografts and associated mesenteries were excised, weighted and imaged.

*In vivo toxicity.* For histology analysis, 24 h after the last intravenous injection of drugs, major organs (heart, liver, spleen, lungs and kidney) and tumour tissues of the mice were fixed, sectioned and stained with haematoxylin and eosin. The slices were observed by microscope (Leica DMI 6000B). At 24 h after the last intravenous injection of drugs, blood was drawn from the venous plexus of the eyes of the mice. Blood samples were immediately centrifuged at 3,000$g$ for 5 min at 4 °C, and the supernatant was collected for haematological analysis. For blood biochemical analysis, ALT, AST, blood urea nitrogen and creatinine values were measured using Modular analytics (Roche, Germany), as indicators of hepatic (ALT and AST) and renal (blood urea nitrogen and creatinine) functions, respectively. For immunotoxicity assay, the cytokines in serum were determined by using enzyme-linked immunosorbent assay kits for IL-6 and TNF-α (Abcam).

*5′-RACE analysis.* We used 5′-RACE to confirm the RNAi-mediated mechanism of action with *NGF* siRNA as previously reported[58]. 5′-RACE was performed using the GeneRacer kit (Invitrogen) with the manufacturer's instructions. In brief, the total RNA was extracted from the treated tumours and reverse transcribed using a gene-specific primer (*NGF* GSP: 5′- GCTTCCAAAAATTTAATAAAT-3′). The original RNA was removed from the duplex followed by purification using a SNAP column. To PCR-amplify the specific cleavage product, a homopolymetric tail was added to the 3′-end of the cDNA, and two rounds of PCR amplification were performed using *NGF* GSP1 (5′-CCATGCAGTCCTTATAATTTAAAATAATT TAC-3′) and nested with PCR with *NGF* GSP2 (5′-AATTTACAGGTTGAGGTAG GGAGG-3′), which allowed amplification of unknown sequences between the GSP2 and the 5′-end of the mRNA. PCR products were analyzed on 2% agarose gel. Uncropped gel was shown in Supplementary Fig. 19.

*IHC and IF staining.* We used IHC and IF staining to evaluate the level of intratumoral NGF protein and neurites. Tumour tissues were cryo-sectioned into 10 µm slices, fixed with 4% PFA, permeabilized with 0.3% Triton X-100 and blocked by 10% goat serum. Sections were incubated with primary antibody, including rabbit monoclonal antibody against NGF (Abcam, Ab52918, 1:200) and chicken polyclonal antibody against neurofilament (Abcam, Ab72997, 1:200), at 4 °C overnight. For IF, the samples were incubated with secondary antibody conjugated with Alexa Fluor 555 (Invitrogen, A-21429 or A-21437, 1:200) at 37 °C for 1 h. Cell nuclei were counterstained with 4,6-diamidino-2-phenylindole at room temperature for 10 min. The samples were mounted with antifade reagent and observed with confocal microscopy (Carl Zeiss LSM710). For IHC of NGF, the samples were incubated with horseradish peroxidase-linked goat anti-rabbit secondary antibody (Invitrogen, G-21234, 1:500) at room temperature for 30 min, and applied with 3,3′-diaminobenzidine (DAB) substrate solution for 3 min. The slices were counterstained with haematoxylin for 2 min, mounted and observed under optical microscopy (Leica DM4000 B). The neural structures were reconstructed by Imaris 7.2 software from the confocal images of neurofilament (Bitplane). The intratumoral neurite density was evaluated by counting the neurite areas positive to neurofilament antibody per field within the section of neurofilament staining.

*Statistical analysis.* Data were presented as mean ± s.d. The *in vitro* experiments were performed in three independent experiments with at least three repetitions for each condition. The *in vivo* experiments were performed with 4–9 mice for each group. Statistical analysis of the samples was performed using Student's *t*-test, and *P* value of <0.05 was considered significant.

*Data availability.* The data that support the finding of this study are available from the corresponding author on reasonable request.

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

## Acknowledgements

We thank Professor Jane Y. Wu at Northwestern University for critical suggestion and insightful discussions. We thank the Center for Biological Imaging (CBI), Institute of Biophysics, Chinese Academy of Science for help of taking the cryoTEM images. This work is supported by CAS (XDA09030305), CAS/SAFEA International Partnership Program for Creative Research Teams, NSFC (81361140345, 31470911, 21535001 and 81673039). Y.F.L. is supported by the Scientific Research Foundation for the Returned Overseas Chinese Scholars, State Education Ministry.

## Author contributions

Y.L., W.Z. and X.J. conceived the study; Y.L. and L.T. performed the experiments; Y. Xie and Y. Xianyu helped the preparation of nanomaterials; L.Z., P.W. and Y.H. assisted the animal studies; K.J. helped providing the human pancreatic tumours; Y.H. was involved in discussion; Y.L. and W.Z. analysed and interpreted the data; Y.L., W.Z. and X.J. wrote the manuscript. All the authors commented on the manuscript.

## Additional information

**Competing interests:** The authors declare no competing financial interests.

