## [Peer Review File · Nature Communications]

Reviewers' comments:

Reviewer #1 (Remarks to the Author):

In their current study, Lei et al. performed the pharmacological characterization of a gold-nanoparticle(GNC)-coupled siRNA against nerve growth factor (NGF) for the treatment of pancreatic ductal adenocarcinoma (PDAC). The GNC core structure was composed of an ion cloud (glutathione and oligoarginine) that carries amine-derived positive charge and thus enables a high-density of siRNA shell, protecting it from nucleases. Via in vitro techniques (cytotoxicity assays, locomotion/migration assays, transmission electron microscopy), and in vivo models (subcutaneous, orthotopic, and patient-derived xenograft/PDX implantation), they provided evidence for enhanced tumor tissue penetration, increased resistance to degradation, increased lysosomal escape of GNC-siRNA against NGF when compared to free circulating siRNA. Furthermore, the experiments pointed out in inhibitory effect of NGF on tumor cell proliferation/viability in vitro, and on tumor size, neurite density, and peritoneal metastasis formation in vivo.

Overall, the authors explored an interesting NGF-targeting strategy for the treatment of PDAC. Due to the overexpression of NGF in PDAC (yet just like in many human solid tumors), this strategy may be worth investigating. However, as it currently stands, the study contains too many drawbacks, raises concerns with regard to certain results, as delineated below:

1. In the abstract, the authors imply that the siRNA against NGF would target the nervous microenvironment in PDAC. However, in human PDAC tissues, the leading source of NGF was reported to be cancer cells themselves (Zhu et al. 1999 and 2001).
2. On supplementary figure 1, one can hardly discern any tumor tissue; the captions do not look representative, especially for normal pancreas. Furthermore, the western blot images on supp. Fig. 1A seem to have been subject to pronounced image processing.
3. On suppl. Fig.1C, the neurofilament staining of the tumor tissue does seem to correspond to any neural structure; rather, these also appear to be cancer cells.
4. The results should be replicated with another potent siRNA sequence against NGF (e.g. siRNA no.1), since only one sequence was used in the whole study.
5. How do the authors explain the escape of the GNC-siRNA from the lysosomes? What is the escape rate of GNC-siRNA in comparison to free siRNA from lysosomes?
6. Which side/adverse effects of a GNC-siRNA be expected in vivo? These have not been mentioned or discussed at all in the manuscript. To what extent is the prolonged circulation and uptake of GNC-siRNA cluster toxic to the kidney and the liver?
7. On Figure 3, the used free siRNA resulted in a mere 17% inhibition. This free siRNA sequence can thus not be representative for the majority of the even commercially available, validated siRNA oligonucleotides.
8. On Figure 4, the authors should quantify the extent of gap closure in the migration assay.
9. In the in vitro migration assay, the authors obviously did not use any mitotic inhibitor. Thus, the observed results may simply be due to inhibition of cancer cell proliferation.
10. In the study of the anti-tumor effects in the subcutaneous model, the authors quite artificially injected the GNC-siRNA or the control siRNA as close to the tumor as possible. Having seen the high plasma stability of GNC-siRNA, this tumor-close injection of siRNA is not understandable, as stated, very artificial. Even if this approach is to complement the results of the systematic administration in the orthotopic model, the provided images in Figure 6B show much greater difference between the free and GNC-bound siRNA groups than shown on the graph on Figure 6A.
11. The representative images on Figure 6F with regard to neurofilament staining are not closeup images and do not allow any conclusion on the neurite density. Moreover, on these images, looking at the DAPI stained (obviously mostly cancer cells), there are no differences in the density of cancer cells between the treatment groups.
12. The same concern also holds true for the unconvincing images provided on figure 7K-L.
13. In the section with the orthotopic model, the authors state that lack of weight loss stands for lack of systemic toxic effects. This is an invalid generalization, since systemic effects of liver and

kidney can even lead to weight gain, and animals that recover from tumors (as expected in the GNC-siRNA group) may even have greater weight than the other treatment groups.

14. The "off-target" effects of GNC-nsRNA that the authors propose on page 13 for Figure 7E&H are not visible on the provided graphs since the tumor weight and tumor size are completely equal on figure 7F&H.

15. From how many different patients were the PDX-models generated?

16. It is laudable that the authors used PDX-models, but PDX do still overcome the most important problem for all PDAC models, i.e. the pronounced stroma that may inhibit siRNA uptake. All the models that the authors used in the present are interesting for the initial pharmacological exploratory phase, yet these models and the obtained results do not efficiently reflect the biology of the human disease as e.g. genetically engineered mouse models like the KPC model of PDAC. Therefore, the true bio-clinical relevance of the demonstrated results, together with the above mentioned concerns, remains very questionable.

17. Although the authors wrote a combined results&discussion section, this section does not contain a thoughtful or insightful discussion of the results in the light of the current literature. Especially, a comparison with other nano-particle delivery strategies, discussion of the caveats/weaknesses of the study, and more insight into future directions would have been very useful. As it currently stands, the study lacks any useful discussion.

Reviewer #2 (Remarks to the Author):

The manuscript by Lei et al. describes the preparation, characterization and functional evaluation of gold nanoclusters for therapeutic siRNA delivery to pancreatic tumor cells in vitro and in vivo. The authors show that GNCs stabilize siRNA in serum and assist in siRNA uptake by tumor cells, resulting in downregulation of the target gene in vitro as well as in vivo. Importantly, the authors show therapeutic benefit of their treatment in three pancreatic tumor models in mice.

The manuscript is well written, methods well designed and described and conclusions are sound. This manuscript is certainly of potential interest for the readers of Nature Communication in the fields of drug delivery as well as (pancreatic) cancer therapy.

Pancreatic cancer is one of the deadline human cancers, and therapeutic options are limited. This manuscript is therefore highly timely. The authors chose to target the nervous microenvironment via downregulation of nerve growth factor, which is an approach that is relatively novel. Nevertheless, two reports have previously shown therapeutic benefit of blocking NGF signaling in pancreatic cancer (PMID 10473107 and 10415871), and NGF siRNA has previously been shown to inhibit growth of breast cancer xenografts (PMID 18199526). This somewhat limits the novelty of this manuscript.

Furthermore, despite the interesting findings, there are several important points that, before publication is considered, should be addressed and/or further evidence should be provided:

Major comments:

- The authors claim in the abstract that 'no report has used nanomaterials to target the nervous microenvironment to suppress pancreatic cancer'. This is not entirely true, as the same authors have previously shown that nanoparticles loaded with neuronal drugs also impair pancreatic tumor progression (PMID: 27046157).
- Figure 1C / supplementary table 4. The authors show that upon adsorption of siRNA onto cationic GNCs, the hydrodynamic diameter of the particles increases from 2.66 nm to 70 nm. This is not reflected in the scheme presented in figure 1C. Why does the particle size increase? Do the gold nanoclusters form aggregates? TEM images of GNC:siRNA complexes should be included here.
- Figure 3A. Gel retardation experiments (Electrophoretic mobility shift assay, EMSA) are also

being used in the field to show successful siRNA complexation to carriers. In figure 3A, it seems that the siRNA is not in complex with GNCs after incubation with serum, as no retardation is observed. One would expect the mobility of the siRNA to be reduced when complexed to GNCs. In addition: Why was the 360 min sample taken from a different gel?

- The authors claim that free siRNA shows target gene inhibition in vitro at 100 nM (figure 3D,E), in vivo after peritumoral injections (4 ug / injection) (figure 6D,E) and even after systemic injections (30 ug/injection) (figure 8 H,I). This also resulted in tumor growth inhibition (Figure 6C and 8F). It is well known, with hundreds of examples in literature, that free, unmodified siRNA is incapable of passing cellular membranes and induce RNA interference. This raises questions on the mechanism through which NGF is downregulated in this study. To confirm knockdown of mRNA through RNAi in vivo, both for the free siRNA and the GNC-siRNA groups, the authors should perform 5'RACE experiments (PMDI 19942683).

- Figure 5. Surprisingly, the authors observe significant amounts of free siRNA (8%) still present in the circulation 1h after injection. As unmodified siRNA has been shown to be degraded/cleared within 1 minute after systemic injection (see e.g. PMID 19401674), the authors may be looking at the kinetics of the fluorescent label instead of the siRNA. For this reason, circulation time and tumor targeting studies should also be performed using labeled GNCs, to confirm delivery of intact complexes to the tumor site.

In addition, signals from other organs (liver, spleen, kidney, lungs etc) should be included in the analysis, especially since in figure 5B,C the majority of the signal seems to be coming from the tumor (also for naked siRNA) which is strange, especially considering the time frame (6h after injection which seems short for EPR effect to take place).

- Figure 8 lacks crucial controls: GNCs complexed with non-specific siRNA. I believe these controls are important in any RNAi experiment.

Minor comments:

- Results 2.4 "we examined the internalization of different siRNA formulations". In figure 3B/C, only one formulation is shown.

- Figure 3C: Multiple, representative cells or groups of cells should be shown in this figure. Ideally, colocalization should be quantified over time.

- Figure 4A,B. The same assay has been used by the authors to evaluate cytotoxicity and proliferation of PANC-1 cells. Still, at 100 nM siRNA, the authors do not observe any cytotoxicity of their formulation after 24h (A), while proliferation seems already significantly inhibited after 24h (B, compared blue to red/black line). Can the authors explain this discrepancy?

- Figure 4D,E. Are the authors looking at an effect on migration here or at an effect on proliferation?

Reviewers' comments:

Reviewer #1 (Remarks to the Author)

In their current study, Lei et al. performed the pharmacological characterization of a gold-nanoparticle(GNC)-coupled siRNA against nerve growth factor (NGF) for the treatment of pancreatic ductal adenocarcinoma (PDAC). The GNC core structure was composed of an ion cloud (glutathione and oligoarginine) that carries amine-derived positive charge and thus enables a high-density of siRNA shell, protecting it from nucleases. Via in vitro techniques (cytotoxicity assays, locomotion/migration assays, transmission electron microscopy), and in vivo models (subcutaneous, orthotopic, and patient-derived xenograft/PDX implantation), they provided evidence for enhanced tumor tissue penetration, increased resistance to degradation, increased lysosomal escape of GNC-siRNA against NGF when compared to free circulating siRNA. Furthermore, the experiments pointed out in inhibitory effect of NGF on tumor cell proliferation/viability in vitro, and on tumor size, neurite density, and peritoneal metastasis formation in vivo.

Overall, the authors explored an interesting NGF-targeting strategy for the treatment of PDAC. Due to the overexpression of NGF in PDAC (yet just like **in many human solid tumors**), **this strategy may be worth investigating**. However, as it currently stands, the study contains too many drawbacks, raises concerns with regard to certain results, as delineated below:

Response: Thank you very much for your evaluation on our manuscript. In this work, we targeted the NGF gene in pancreatic cancer. As the reviewer indicated, NGF is also overexpressed in many other solid tumors, thus we believe that our strategy could be broadened to other types of cancers, which may give a new therapeutic direction for pancreatic cancer and possible other cancers.

We also thank the reviewer for giving us the critical comments. Please find in the attached point-by-point response to the reviewer's comments.

1. In the abstract, the authors imply that the siRNA against NGF would target the nervous microenvironment in PDAC. However, in human PDAC tissues, the leading source of NGF was reported to **be cancer cells themselves** (Zhu et al. 1999 and 2001).

Response: Thank you very much for pointing out this critical concern. Yes, it is true that the cancer cells themselves in human PDAC tissues secrete NGF (Suppl. Fig. 1). So that is exactly the reason/concept in our study that we aim to silence the NGF gene in pancreatic cancer cells/pancreatic tumors. Please find the updated Fig. 10 for the concept of using GNC-siRNA for NGF silencing and pancreatic cancer treatment. In this modification, we show the position of cancer cell membrane, to separate the cancer cells and the extracellular microenvironment. The NGF knockdown in pancreatic cancer cells will inhibit both pancreatic cancer growth and neuron growth in the tumor microenvironment.

Figure 10

2. On supplementary figure 1, one can hardly discern any tumor tissue; the captions do not look representative, especially for normal pancreas. Furthermore, the western blot images on supp. Fig. 1A seem to have been subject to pronounced image processing.

Response: Thank you very much for pointing out these critical concerns.

In fact, we have isolated the tumor tissues (solid and hard tissue) from the normal pancreas tissues (soft tissue), and then performed the staining and analysis. So we can indeed compare the two parts (Suppl. Fig. 1).

For the western blot in suppl. Fig. 1A, in previous version of manuscript, we used the WB images with minimum background. In this version, we redid the WB and used the WB results with less reduced background. And we consistently made this kind of change throughout the revised manuscript.

Suppl. Figure 1

3. On suppl. Fig. 1C, the neurofilament staining of the tumor tissue does seem to correspond to any neural structure; rather, these also appear to be cancer cells.

Response: Thank you very much for pointing out this critical concern.

As suggested, we redid the neurofilament staining in both pancreatic tumor biopsy and in the normal pancreas tissues without development of tumors. Moreover, we did the reconstruction of neural structure using Imaris 7.2 software, and the updated results were shown in suppl. Fig. 1C (as shown above). By doing this, we can clearly demonstrate that there are more abundant neurite formation in pancreatic tumors compared to the regular pancreas tissues.

4. The results should be replicated with another potent siRNA sequence against NGF (e.g. siRNA no.1), since only one sequence was used in the whole study.

Response: Thank you very much for pointing out this critical concern. In fact, we have synthesized 5 siRNA sequences against NGF (suppl. Table 1), and screened their effects in down-regulation of NGF. We selected the one with the greatest knockdown efficiency of NGF mRNA (siRNA#2, suppl. Fig. 2) to carry out the following experiments. Therefore, the sequence we used represents the most potent effect against NGF in this study.

Suppl. Figure 2

5. How do the authors explain the escape of the GNC-siRNA from the lysosomes? What is the escape rate of GNC-siRNA in comparison to free siRNA from lysosomes?

Response: Thank for pointing out these critical questions.

In fact, two prerequisites are required for efficient siRNA-mediated gene silencing: (1) the high level of cellular uptake of siRNA; (2) the successful release of siRNA into the cytoplasm. From Fig. 3B, we proved that the Panc-1 cells had a significant uptake of GNC-siRNA complex compared to free siRNA; from Fig. 3C, we showed that the GNC-siRNA complex is able to enter the lysosomes first (1 h), then escape from the lysosomes into the cytoplasm of the cells (4 h). These two points enable the efficient siRNA-mediated gene silencing by GNC-siRNA complex. Considering the complexity of the GNC-siRNA complex and lysosome interaction, the exact mechanism of the lysosome escape of the GNC-siRNA complex will be studied in our future work.

Previously, we found that the free siRNA hardly entered the cells (Fig. 3B), which is due to its high molecular weight and high density of negative charge. Since almost no free siRNA can enter the cells, it was hard to evaluate its escape from lysosome (Supplementary Fig. 12). Thus, it is also very hard to compare the lysosome-escape rates of GNC-siRNA complex and free siRNA.

Figure 3

Suppl. Figure 12

However, as the reviewer suggested, we first quantified the colocalization of GNC-siRNA with lysosomes over time (Supplementary Fig. 11A), which was consistent with the observation in Fig. 3C. The result also showed a maximum colocalization of GNC-siRNA in lysosomes at 2 h. We observed that the GNC-siRNA began to release from the lysosomes at 2 h, so we measured the escape ratio of GNC-siRNA from the lysosomes from 2 h to 6 h (Supplementary Fig. 11B), the escape ratio of GNC-siRNA from lysosomes increased from 40% to 76% from 3 h to 6 h. We updated this result in Page 8 of the revised manuscript and in Page 12 of the revised supporting information.

Suppl. Figure 11

6. Which side/adverse effects of a GNC-siRNA are expected *in vivo*? These have not been mentioned or discussed at all in the manuscript. To what extent is the prolonged circulation and uptake of GNC-siRNA cluster toxic to the kidney and the liver?

Response: Thank you very much for pointing out your critical concerns. As you suggested, we evaluated *in vivo* toxicity of the GNC-siRNA complex. The experiments and results were added into the revised manuscript (Page 15-16, Page 26, and Page 44).

24 h after the final i.v. injection of drugs, histopathology of different organs and tumors were evaluated by H&E staining. From the H&E-stained tissues sections of heart, liver, spleen, lung and kidney, no obvious histological changes were observed between the controls and GNC-siRNA group (Fig. 9A). However, many tumor cells were dead in the GNC-siRNA group (Fig. 9A), indicating the anti-tumor efficacy with minimal organ toxicity from GNC-siRNA complex.

In addition, 24 h after the last i.v. injection of the drugs, we collected the blood samples for hematological analysis. From biochemical analysis, GNC-siRNA group kept the parameters of liver and kidney functions within the normal range (Fig. 9B). However, free siRNA group induced a relatively high level of ALT and AST (Fig. 9B), indicating hepatic dysfunction from the free siRNA group. At the therapeutic doses, no significant immune response was induced by GNC-siRNA complex, as the production of serum cytokines, such as interleukin-6 (IL-6) and tumor necrosis factor- α (TNF- α) was similar to saline control group (Fig. 9C). However, the free siRNA group induced relatively high immune responses by production of more IL-6 and TNF- α (Fig. 9C). Together, GNC-siRNA complex showed safety and low toxicity *in vivo* compared to free siRNA formulation.

Figure 9

7. On Figure 3, the used free siRNA resulted in a mere 17% inhibition. This free siRNA sequence can thus not be representative for the majority of the even commercially available, validated siRNA oligonucleotides.

Response: Thank you for pointing out this critical concern. We feel that the reviewer may misunderstand our results.

In fact, researchers generally use the commercially available transfection agents (such as Lipofectamine® 2000 from Invitrogen, ViaFect™ Transfection Reagent from Promega, and so on). In our study, we previously screened a series of nanomaterials-siRNA conjugates for the greatest knockdown of NGF, and we have indeed used Lipofectamine® 2000 transfection agent to delivery siRNA as a positive control (Lipo2000-siRNA) (Suppl. Fig. 4), the red dotted line in the Suppl. Fig. 4 referred to the expression level of NGF mRNA in cells transfected with Lipofectamine® 2000, it is about 64% inhibition.

Suppl. Figure 4

In Fig. 3D, the “free siRNA” means the naked siRNA neither with any chemical modification nor with any transfection agent, this free siRNA hardly enter the cells to induce effective siRNA silencing (only 17% inhibition). As you suggested, we added the results of the siRNA transfected with Lipofectamine® 2000 in Fig. 3D for a positive control for clarification (with dotted red line), and we updated the figure and caption accordingly (Page 34-35 of the revised manuscript).

Figure 3D

8. On Figure 4, the authors should quantify the extent of gap closure in the migration assay.

Response: Thank you for your suggestion. As suggested, we calculated the surface area covered by migrating cells in the gap regions, in order to quantify the extent of gap closure in the migration assay. And we have modified the manuscript and figure accordingly (Fig. 4E, Page 9, Page 36 of the revised manuscript).

Figure 4E

9. In the *in vitro* migration assay, the authors obviously did not use any mitotic inhibitor. Thus, the observed results may simply be due to inhibition of cancer cell proliferation.

Response: Thank you for pointing out this critical concern.

The migration assay on microfluidic chips was an original tool developed in our labs (PMID 17183592, PMID 19787666).

Thank you for your suggestions, as suggested, we redid the migration assay with pretreatment of Panc-1 cells with mitotic inhibitor (mitomycin C) prior to the migration assay (Page 23 in the revised manuscript). However, the use of mitotic inhibitor to Panc-1 cells did not influence the migration results, and we obtained the similar tendency as previously (updated Fig. 4D,E, Page 36 of the revised manuscript), that the GNC-siRNA complex inhibited the migration of Panc-1 cells compared to nontreated control and GNC-nsRNA treatment.

Figure 4D&E.

10. In the study of the anti-tumor effects in the subcutaneous model, the authors quite artificially injected the GNC-siRNA or the control siRNA as close to the tumor as possible. Having seen the high plasma stability of GNC-siRNA, this tumor-close injection of siRNA is not understandable, as stated, very artificial. Even if this approach is to complement the results of the systematic administration in the orthotopic model, the provided images in Figure 6B show much greater difference between the free and GNC-bound siRNA groups than shown on the graph on Figure 6A.

Response: Thank you for pointing out these critical concerns.

In the subcutaneous tumor model, we used peritumoral injection for siRNA delivery, in order to evaluate their effect in local administration (thus to avoid any loss in blood circulation). In the orthotopic tumor model, we used intravenous injection via tail vein, in order to evaluate their effect in systemic administration (in blood circulation). By doing this, we can mimic different tumor models and compare different administration methods of the drugs. In both administration methods, the GNC-siRNA shows efficient NGF knockdown and anti-tumor efficacy. However, if the reviewer feels the peritumoral injection in subcutaneous model is not understandable, we could move these results into supporting information or remove these results.

For the comparison of Fig. 6A and Fig. 6B, we feel that the reviewer may misunderstand the orders in figures. To clarify this point, now we use different colors for different groups, and we consistently change all the colors in the figures throughout the manuscript for clarification (Saline-black, Free siRNA-red, GNC-siRNA-blue, GNC-nsRNA-green). In fact, in both Fig. 6A and Fig. 6B, the orders of tumor sizes are: Saline ~ GNC-nsRNA > Free siRNA > GNC-siRNA (where GNC-siRNA means GNCs binding with NGF siRNA, GNC-nsRNA means GNCs binding with nonsense siRNA). The results were quite consistent from one with another.

Figure 6A&B

11. The representative images on Figure 6F with regard to neurofilament staining are not closeup images and do not allow any conclusion on the neurite density. Moreover, on these images, looking at the DAPI stained (obviously mostly cancer cells), there are no differences in the density of cancer cells between the treatment groups.

Response: Thank you for pointing out these critical concerns.

As you suggested, we put both low-magnification and high-magnification images in the updated Fig. 6F, to clearly show the neural staining in different groups, and we made this kind of change for all the neurite staining images throughout the revised manuscript. Moreover, we have quantified the density of neurites in the tumor sections, as shown in Fig. 6G.

Figure 6F

Figure 6G

Concerning the DAPI staining in Fig.6F, we need to use high-magnification images to show the neural signal. Within high magnified images, the difference of the tumor cell density in various groups was not so obvious. However, as you suggested, we performed H&E staining on (orthotopic) pancreatic tumor sections (right column in Fig. 9A). Within these low magnified images, we found that many tumor cells were dead in the GNC-siRNA group (less blue color), indicating the anti-tumor efficacy from the GNC-siRNA group.

Figure 9A

12. The same concern also holds true for the unconvincing images provided on figure 7K-L.

Response: Similarly to the responses to the above question, we have added the high-magnification images in Fig. 7. In fact, Fig. 7J showed the WB result of NGF protein level in tumor tissues, Fig. 7L corresponded to the NGF staining in tumor tissue. Fig. 7M illustrated the neural structure in tumor tissues by staining, and Fig. 7N showed the quantification of neurite density in tumor tissues.

Figure 7

13. In the section with the orthotopic model, the authors state that lack of weight loss stands for lack of systemic toxic effects. This is an invalid generalization, since systemic effects of liver and kidney can even lead to weight gain, and animals that recover from tumors (as expected in the GNC-siRNA group) may even have greater weight than the other treatment groups.

Response: Thank you for pointing out this critical concern. As the reviewer suggested, we did not only use the mouse weight as a parameter to evaluate the systematic toxic effects in the revised manuscript, but also carried out the H&E staining of different organs and hematological assays to evaluate the systematic toxicity of the drugs. We updated these results in new Fig. 9 in the revised manuscript. Please refer to the details in the response to question 6 above.

14. The "off-target" effects of GNC-nsRNA that the authors propose on page 13 for Figure

7E&H are not visible on the provided graphs since the tumor weight and tumor size are completely equal on figure 7F&H. (should be “7E&H”)

Response: We think we misused the concept of “off-target” in our previous manuscript, so we will not use this concept in the revised manuscript. As the reviewer suggested, we modified the description on these results, now we described that “The GNC-nsRNA did not show a significant inhibitory effect on tumor growth”, and we deleted the sentence “The minor anti-tumor effects observed for nonsense siRNA (GNC-nsRNA) might be caused by the “off-target” effect of siRNA as reported previously” for clarification (Page 13 of the revised manuscript).

Also, we feel that the reviewer may indicate that the 7E&H are equal. For clarity, we used different colors for different groups (saline-black, Free siRNA-red, GNC-siRNA-blue, GNC-nsRNA-green) in the revised manuscript. In fact, Fig. 7F corresponded to Fig. 7C, they were the results from the bioluminescence signal from the Panc-1-luc tumors in situ. Fig. 7H corresponded to Fig. 7E, they were the image and the weight of the isolated tumors ex vivo. The two quantification methods are different.

Figure 7

15. From how many different patients were the PDX-models generated?

Response: We used the tumors from five different patients with pancreatic cancer. For each patient’s tumor, we did a series of experiments to compare the effect of different siRNA formulations. Now we clarified this statement in Page 26 in the revised manuscript.

16. It is laudable that the authors used PDX-models, but PDX do still overcome the most important problem for all PDAC models, i.e. the pronounced stroma that may inhibit siRNA uptake. All the models that the authors used in the present are interesting for the initial pharmacological exploratory phase, yet these models and the obtained results do not efficiently reflect the biology of the human disease as e.g. genetically engineered mouse

models like the KPC model of PDAC. Therefore, the true bio-clinical relevance of the demonstrated results, together with the above mentioned concerns, remains very questionable.

Response: Thanks for your nice comments and kind suggestions.

We thank the reviewer for the suggestion to use KPC model mice (in details, which harbor heterozygous mutant alleles of Kras, p53 and a pancreatic-specific Cre recombinase, Pdx1-Cre; namely, $Kras^{LSL-G12D/+}$, $Trp53^{LSL-R172H/+}$, Pdx-1 Cre) for pancreatic cancer model. However, we do not have these KPC mice at the moment in China, and it takes at least nine month to generate this mouse model. So, at the moment, we did not use the KPC model in this study. We will consider using these mice in our future studies.

In this study, we used the patient-derived xenograft (PDX) tumor model, based on follow considerations: first, the PDX tumor model is a preclinical model and gives great opportunities in oncology and drug development (PMID 22508028). Second, this work was in collaboration with the Department of Hepatobiliary Surgery, Chinese PLA General Hospital, China. We are able to develop our study using human patients' tumors, in turn, our results will provide effective feedbacks to the treatment of pancreatic cancer. As the reviewer also mentioned, the PDX model is interesting for the initial pharmacological exploratory phase, our present work is only in this phase, not for the clinical drug evaluation yet.

17. Although the authors wrote a combined results & discussion section, this section does not contain a thoughtful or insightful discussion of the results in the light of the current literature. Especially, a comparison with other nano-particle delivery strategies, discussion of the caveats/weaknesses of the study, and more insight into future directions would have been very useful. As it currently stands, the study lacks any useful discussion.

Response: Thank you for giving us these critical comments.

As suggested, first, we separated the results and discussion into two sections. In the new section of discussion (Page 16-18 in the revised manuscript), first, we clarified the novelty of our study, namely using the new class of materials (GNCs), and targeting the NGF in pancreatic cancer. Second, we compared our strategies with the current techniques in the literature. We emphasized that the GNCs had the highest loading capacity of siRNA (GNC-siRNA) up to date, which was also the reason that the GNC-siRNA complex resulted in the most effective gene knockdown effect. Third, we investigated the mechanism of targeting the nervous microenvironment *via* NGF silencing for pancreatic cancer therapy (Fig. 10). In the end, we discussed on the weaknesses of the present study, and the perspective works in the future (Page 18 of the revised manuscript).

Reviewer #2 (Remarks to the Author):

The manuscript by Lei et al. describes the preparation, characterization and functional evaluation of gold nanoclusters for therapeutic siRNA delivery to pancreatic tumor cells in vitro and in vivo. The authors show that GNCs stabilize siRNA in serum and assist in siRNA uptake by tumor cells, resulting in down regulation of the target gene in vitro as well as in vivo. Importantly, the authors show therapeutic benefit of their treatment in three pancreatic tumor models in mice.

The manuscript is **well written**, methods well designed and described and conclusions are sound. This manuscript is certainly of potential interest for the readers of Nature Communication in the fields of drug delivery as well as (pancreatic) cancer therapy.

Pancreatic cancer is one of the deadline human cancers, and therapeutic options are limited. This manuscript is therefore **highly timely**. The authors chose to target the nervous microenvironment via down regulation of nerve growth factor, which is an approach that is **relatively novel**. Nevertheless, two reports have previously shown therapeutic benefit of blocking NGF signaling in pancreatic cancer (PMID 10473107 and 10415871), and NGF siRNA has previously been shown to inhibit growth of breast cancer xenografts (PMID 18199526). This somewhat limits the novelty of this manuscript.

Response: Thank you very much for your very positive assessment to our manuscript. All your comments help us to greatly improve the quality of the manuscript.

Thank you for pointing out these related publications to us. In PMID 10473107 and 10415871, both papers showed the use of Trk tyrosine kinase inhibitor (CEP-701) to treat pancreatic cancer. Noticing that Trk(A) is NGF receptor, these studies implied that target NGF may be also useful for pancreatic cancer. Although targeting the same pathway, in our study, more directly, we managed to suppress the produce of NGF which is on the upstream of the Trk(A) signaling and may be more efficient in suppressing NGF-mediated downstream signaling pathways. The direct targeting to NGF production for pancreatic cancer therapy has not yet been reported in the previous literature. Thus, our strategy is novel compared to published studies. Furthermore, we silenced NGF in pancreatic tumors using NGF siRNA which is also not reported previously. Moreover, we used GNCs to enhance the delivery of NGF siRNA and achieved unprecedented siRNA loading capacity and excellent pancreatic cancer inhibition (The details shown below).

PMID 18199526 is the first paper which showed the use of NGF siRNA for (breast) cancer treatment. This paper only used subcutaneous tumor model, and used the peritumoral injection of siRNA in naked siRNA formulation. In our study, we developed gold nanoclusters for efficient delivery of NGF siRNA (GNC-siRNA). The GNC-siRNA

complex enhanced the stability and efficacy of siRNA compared to free siRNA formulation, and showed effective anti-tumor efficiency in three tumor models (subcutaneous, orthotopic, and PDX models), including both local and systemic administration.

Thus, our paper provides new insights into the pancreatic cancer therapy.

Additionally, we clarify the **novelty** of our study, as listed below:

1) Use of the gold nanoclusters for efficient delivery of NGF siRNA.

We used gold nanoclusters for efficient delivering of NGF siRNA (GNG-siRNA), Gold nanoclusters with diameters less than 3 nm are a novel class of gold materials. The GNCs developed in this study had an unprecedented loading capacity for siRNA (**226 $\mu\text{mol siRNA/g GNCs}$** , results in Table S3). To our knowledge, the GNCs had the highest loading of siRNA compared to the previous works (reviews in PMID 25057313, 19180106).

Previous researchers developed 13-nm-diameter gold nanoparticles (GNPs) for delivery of siRNA in several disease model (PMID 19170493, 24174328), the GNPs had a loading capacity of ~ 40 siRNA stands per particle (equal to **3 $\mu\text{mol siRNA/g GNPs}$**). Thus, the GNCs in our study provided a much higher loading capacity of siRNA compared to the actually existed technology, most likely due to the smaller size and larger specific surface area of the GNCs and the positive charge on the surface of GNCs.

Moreover, due to the high loading capacity of siRNA, the GNC-siRNA complex resulted in the greatest gene knockdown among a series of nanomaterials-siRNA conjugates with the same concentration of siRNA screened in this study (Table S3).

The GNC-siRNA complex stabilized the siRNA in the blood stream, enhanced the siRNA uptake by cancer cells and tumor tissues, and potently down-regulated the target gene in pancreatic cancer cells and in pancreatic tumor tissues, and significantly inhibited tumor growth in three kinds of pancreatic tumor models.

Thus, our work provided a straightforward but very effective approach for gene silencing, resulting in the inhibition of pancreatic cancer progression via NGF silencing.

2). Target the nervous microenvironment in pancreatic cancer.

Pancreatic cancer is one of the deadliest human cancers, but therapeutic options are currently limited. Thus, the development of new therapeutic approach is urgently needed.

In this study, we proposed strategy to target the nervous microenvironment in PDAC based on follow considerations: the nervous microenvironment has been recently recognized as a novel niche which contribute to cancer progression (*Science* 2013, PMID 23846904, on prostate cancer; *Science Translational Medicine* 2014, PMID 25143365, on gastric cancer; *Nature Communications* 2015, PMID 25917569, on head and neck cancer; *Cell* 2015, PMID 25913192, on glioma; *Cell Stem Cell* 2015, PMID 25842978, on skin cancer). More specifically, the nervous microenvironment has a crucial impact on the invasive growth and metastasis of pancreatic cancer, the perineural invasion happens in up

to 100% in PDAC. NGF is over-expressed by pancreatic cancer cells and pancreatic tumors, and contributes to the progression of PDAC.

Based on these observations, we aimed to develop this anti-neurogenic therapy by silencing NGF gene in pancreatic cancer, in order to regulate its progression.

Up to date, no work has used NGF siRNA in PDAC to target the nervous microenvironment and to treat PDAC. Moreover, in this study, we use nanotechnology (GNCs) to greatly enhance the siRNA delivery and treatment efficiency of PDAC.

In this study, we have showed the therapeutic benefit of GNC-siRNA complex in three kinds of pancreatic tumor models, including subcutaneous model, orthotopic model, and patient-derived xenograft (PDX)-model. Thus our study provided effective animal models to taking this treatment into human pancreatic cancer in clinics.

Thus, targeting the nervous microenvironment via NGF silencing in pancreatic cancer, represents the second novelty of this study.

Altogether, these two points, extremely high siRNA delivery capability of the GNCs and successful nervous microenvironment (NGF gene) targeting in pancreatic cancer therapy, comprise the novelty of this paper.

Furthermore, despite the interesting findings, there are several important points that, before publication is considered, should be addressed and/or further evidence should be provided:

Major comments:

- The authors claim in the abstract that 'no report has used nanomaterials to target the nervous microenvironment to suppress pancreatic cancer'. This is not entirely true, as the same authors have previously shown that nanoparticles loaded with neuronal drugs also impair pancreatic tumor progression (PMID: 27046157).

Response: Thank you for pointing out this critical concern.

The above mentioned publication (Lei et al., *J Control Release* 2016, PMID 27046157) is from our group. In the published work, we used ferritin nanoparticles to encapsulate the neuronal drugs, with the targeted release of drugs, which will activate/block the acetylcholine-muscarinic signaling (one of the signaling in nervous microenvironment) in the pancreatic tumor microenvironment.

Since NGF is over-expressed by pancreatic tumors, more directly, in present study, we aimed to silence the NGF gene in pancreatic cancer cells/pancreatic tumors, with gold nanoclusters conjugated with NGF siRNA (GNC-siRNA). By doing these, we can suppress NGF gene and neurogenesis in pancreatic tumors, and inhibit the pancreatic cancer progression (scheme in Fig. 10).

As suggested, we changed the sentence, instead of saying “No report has used nanomaterials to target the nervous microenvironment to suppress pancreatic cancer”, now we use the sentence “The suppression of gene expression of nerve growth factor (NGF)

may have great potential in pancreatic cancer treatment” in the abstract (Page 2 of the revised manuscript).

Figure 10

• Figure 1C / supplementary table 4. The authors show that upon adsorption of siRNA onto cationic GNCs, the hydrodynamic diameter of the particles increases from 2.66 nm to 70 nm. This is not reflected in the scheme presented in figure 1C. Why does the particle size increase? Do the gold nanoclusters form aggregates? TEM images of GNC:siRNA complexes should be included here.

Response: Thank you very much for your critical concern.

As you suggested, we carried out the structure analysis of GNC-siRNA complex by cryoTEM (previously, with normal TEM assay, we can not observe the structure of GNC-siRNA complex, due to the highly different contrast between GNCs (metal) and siRNA (nucleotides). From the cryoTEM analysis, the GNCs and GNC-siRNA complex had a diameter of 2.6 ± 0.5 , and 16.6 ± 3.0 nm, respectively (Fig. 1G,I). Each GNC-siRNA complex contained from 3 to up to 16 GNCs in the structure (Suppl. Fig. 7).

Figure 1G&I

Suppl. Figure 7

As the reviewer suggested, we redraw the scheme in Fig. 1C, by considering the sizes of the GNCs and siRNA: the diameter of GNCs is 2.6 nm, and the siRNA is ~ 7.5 nm in length and 2 nm in diameter (PMID 20059641 review). We kept the size ratio between the GNCs and siRNA, and we also considered the cryoTEM result of the GNC-siRNA complex.

Figure 1C

With DLS assay, the diameter of GNCs and GNC-siRNA were 6.6 ± 0.8 nm and 70.2 ± 8.1 , respectively (suppl. Table 4). The TEM size is generally smaller than DLS size (suppl. Table 4), since TEM gives the dehydrated diameter of samples at the dried state, whereas DLS gives the hydrodynamic diameter of the samples at the hydrated state.

Supplementary Table 4

	Size by cryoTEM (nm)	Hydrodynamic size by DLS (nm)	Polydispersity index (PDI)	Zeta potential (mV)
GNCs	2.6 ± 0.5	6.6 ± 0.8	0.111 ± 0.008	19.9 ± 0.8
GNC-siRNA	16.6 ± 3.0	70.2 ± 8.1	0.224 ± 0.016	-22.6 ± 1.6

In addition, we conducted AFM analysis to show the morphology of GNCs and GNC-siRNA, respectively (Suppl. Fig. 8), AFM analysis also revealed that after binding siRNA with GNCs, the GNC-siRNA complex had a higher height (~ 25.6 nm) than GNCs (~2.2 nm) (Suppl. Fig. 8).

Suppl. Figure 8

• Figure 3A. Gel retardation experiments (Electrophoretic mobility shift assay, EMSA) are also being used in the field to show successful siRNA complexation to carriers. In figure 3A, it seems that the siRNA is not in complex with GNCs after incubation with serum, as no retardation is observed. One would expect the mobility of the siRNA to be reduced when complexed to GNCs.

In addition: Why was the 360 min sample taken from a different gel?

Response: Thank you for pointing out your critical concerns. In fact, previously, in order to save place in the image, we put the two groups (free siRNA and GNC-siRNA, at 0, 15, 30, 45 and 60 min, respectively) at the same height. Previously, we used a 10-well gel, thus we had to put the sample of GNC-siRNA complex at 360 min on another gel.

As you suggested, we redid this experiment with the gel containing 15 wells, this time we were able to show all the samples in the same gel. Indeed, at 0 min, the electrophoretic mobility of GNC-siRNA complex was less than that of the free siRNA (Fig. 3A, at 0 min), indicating the GNC-siRNA was able to form complexes that were stable against the electrophoretic force applied during the electrophoresis. Moreover, we detected the siRNA component via Gel-Red nucleic acid staining after incubation of the free siRNA and GNC-siRNA in the presence of 10% serum for multiple time points (Fig. 3A). GNC-siRNA retarded siRNA degradation in serum condition, and the intensity of siRNA remained over a period of 6 h incubation, indicating that GNCs protect the siRNA in the GNC-siRNA formulation (Fig. 3A). In contrast, free siRNA was rapidly degraded in serum condition, and no free siRNA was detected on the gel (Fig. 3A). This result indicated that GNC-siRNA complex effectively protected the siRNA against serum nuclease degradation. And this information is updated in Page 7 and Page 34 of the revised manuscript.

Figure 3A

• The authors claim that free siRNA shows target gene inhibition *in vitro* at 100 nM (figure 3D,E), *in vivo* after peritumoral injections (4 ug / injection) (figure 6D,E) and even after systemic injections (30 ug/injection) (figure 8 H,I). This also resulted in tumor growth inhibition (Figure 6C and 8F). It is well known, with hundreds of examples in literature, that free, unmodified siRNA is incapable of passing cellular membranes and induce RNA interference. This raises questions on the mechanism through which NGF is down regulated in this study. To confirm knockdown of mRNA through RNAi *in vivo*, both for the free siRNA and the GNC-siRNA groups, the authors should perform 5'RACE experiments (PMDI 19942683: A rapid and sensitive method to detect siRNA-mediated mRNA cleavage *in vivo* using 5' RACE and a molecular beacon probe.).

Response: Thank you for pointing out these critical concerns and your suggestion on 5'RACE assay to clarify the RNAi-mechanism *in vivo* in our study.

As the reviewer suggested (in latter question), we first added the results on GNC-nsRNA group (GNCs binding with nonsense NGF siRNA) (updated Figure 8).

Free siRNA slightly inhibited NGF expression in subcutaneous tumors by peritumoral injection (Fig.6D,E, $0.01 < p < 0.05$), but did not significantly induce NGF inhibition in the *in vitro* test in cells (Fig. 3D,E, $p > 0.05$), nor in the orthotopic tumors by systemic inject (Fig. 7I, J, $p > 0.05$). However, we noticed slight anti-tumor effect of free siRNA in PDX tumor model (Fig. 8F, $0.01 < p < 0.05$; but in Fig.8H, $p > 0.05$). Despite some of “inconsistency” of the results on free siRNA, the GNC-siRNA consistently inhibited the tumor growth and reduced the NGF expression in different models (Fig. 3, Fig. 6-8).

As the reviewer suggested, we performed the 5'RACE assay (on the orthotopic pancreatic tumor samples) to verify the *in vivo* RNAi mechanism. 5'-RACE analysis of total RNA using human NGF gene-specific primers identified a PCR product of expected cleavage product with a molecular weight ~370 bp in the GNC-siRNA-treated tumors (Fig. 7K). This band was not obvious in the free-siRNA and GNC-nsRNA -treated tumors (Fig. 7K). The NGF siRNA-mediated mRNA cleavage products thus confirmed a sequence-specific, RNA-induced silencing complex (RISC) mediated activity in tumors treated with GNC-siRNA complex. And we updated the relevant contents in Page 14, Page 27, and Page 39-41 of the revised manuscript.

Figure 7K

• Figure 5. Surprisingly, the authors observe significant amounts of free siRNA (8%) still present in the circulation 1h after injection. As unmodified siRNA has been shown to be degraded/cleared within 1 minute after systemic injection (see e.g. PMID 19401674), the authors may be looking at the kinetics of the fluorescent label instead of the siRNA. For this reason, circulation time and tumor targeting studies should also be performed using labeled GNCs, to confirm delivery of intact complexes to the tumor site.

Response: Thank you for pointing out this critical concern.

In this study, we need to compare the stability and biodistribution of siRNA (free siRNA or GNC-siRNA formulation), thus previously, we labeled the siRNA with a far-red-fluorescent Cy5 dye (ex/em 649/670 nm). Cy5 dye can diminish the background signal from the blood during observation in the *in vivo* imaging system (Maestro™ 2, CRi).

We have also tried to figure out the *in vivo* distribution of GNCs by fluorescence imaging. However, since the GNCs had ex/em at 430/596 nm, it was difficult to distinguish the signal of GNCs from the background signal of blood in the *in vivo* imaging system.

Now, we notice that the Cy5 dye indeed has a long half-life time in blood, thus as the reviewer supposed, it seems that the signal in kinetic study may partly come from the fluorescent label Cy5 dye.

As the reviewer suggested, we performed the inductively coupled plasma mass spectrometry (ICP-MS) to show the *in vivo* distribution of gold element in the GNC-siRNA complex (Fig. 5E). The result confirmed the accumulation of gold element (in the GNC-siRNA complex) into the tumor regions (Fig. 5E). Both enhanced fluorescence intensity of Cy5-siRNA and gold concentration at the tumor regions confirmed an increased accumulation of the GNC-siRNA to the tumor sites. And we updated the relevant information in Page 11, Page 24-25, and Page 37 of the revised manuscript.

Figure 5

In addition, signals from other organs (liver, spleen, kidney, lungs etc) should be included in the analysis, especially since in figure 5B,C the majority of the signal seems to be coming from the tumor (also for naked siRNA) which is strange, especially considering the time frame (6h after injection which seems short for EPR effect to take place).

Response: Thank you for your suggestion.

As suggested, we have updated the Fig.5B,C, please refer to the above figure for the

results. In the *in vivo* fluorescence imaging assay, we labeled siRNA with Cy5 dye (ex/em at 649/670nm). Fig. 5B showed the fluorescence images of whole mice after 6 h of systemic injection, Fig. 5C showed the biodistribution in major organs and tumors in mice after 24 h of injection.

At 6 h, the GNC-siRNA significantly accumulated into the tumor region with a higher degree than free siRNA and the saline control (Fig. 5B). At 24 h (not at 6 h) following systemic injection, tumors treated with GNC-siRNA exhibited higher level of fluorescent intensity at the tumor sites than those treated with free siRNA (Fig. 5C, D) ($p < 0.01$). Similarly to the above question, it seems the signal in the free siRNA group may partly come from the Cy5 dye.

Thus we also used ICP-MS assay to confirm the accumulation of gold (from GNC-siRNA) into the tumor region. Both enhanced fluorescence intensity of Cy5-siRNA and gold concentration at the tumor regions confirmed an increased accumulation of the GNC-siRNA complex to the tumor sites

• **Figure 8 lacks crucial controls: GNCs complexed with non-specific siRNA. I believe these controls are important in any RNAi experiment.**

Response: Thank you for pointing out this critical concern.

In fact, the results of GNC-nsRNA in the PDX-tumor model were quite consistent with those results in the orthotopic pancreatic tumor model with i.v. injection (Fig. 7), thus previously we did not show these results. However, as the reviewer suggested, now we added the results on GNC-nsRNA group in Fig. 8. And we modified the manuscript accordingly (Page 15, and Page 42-43).

Figure 8

Minor comments:

- Results 2.4 "we examined the internalization of different siRNA formulations". In figure 3B/C, only one formulation is shown.

Response: Thank you for pointing out your critical concern.

In fact, we first examined the internalization of free siRNA and GNC-siRNA complex into the Panc-1 cells (Fig. 3B). GNC-siRNA entered the Panc-1 cells in much larger quantity, the free siRNA hardly entered the cells (Fig. 3B), we also quantified the fluorescence intensity of Cy5-siRNA in the cells for clarification (new Suppl. Fig. 10).

Thus in the main text, we focused only on the lysosomal escape of GNC-siRNA complex (Fig. 3C). However, as the reviewer suggested, we put the results on colocalization of free siRNA and lysosomes in the supporting information (new Suppl. Fig. 12). Since the free siRNA hardly entered the cells (Fig. 3B), it was hard to distinguish the colocalization of free siRNA in lysosomes (Suppl. Fig. 12).

Figure 3

Suppl. Figure 10

Suppl. Figure 12

• **Figure 3C:** Multiple, representative cells or groups of cells should be shown in this figure. Ideally, colocalization should be quantified over time.

Response: Thank you for your suggestions. As suggested, we put more cells in the images, and we put the magnified images of one cell in the right column (Fig. 3C).

As the reviewer suggested, we quantified the colocalization of GNC-siRNA in lysosomes over time (Supplementary Fig. 11A), the result was consistent with the observation in Fig. 3C. And we updated this information in Page 8 of the revised manuscript, and in Page 12 of the revised supporting information.

Suppl. Figure 11

• Figure 4A,B. The same assay has been used by the authors to evaluate cytotoxicity and proliferation of PANC-1 cells. Still, at 100 nM siRNA, the authors do not observe any cytotoxicity of their formulation after 24h (A), while proliferation seems already significantly inhibited after 24h (B, compared blue to red/black line). Can the authors explain this discrepancy?

Response: Thank you for pointing out this concern. We apologize for the unclear description to confuse the reviewer. In fact, the experiment details were different:

In the cytotoxicity assay, Panc-1 cells were plated in 96-well plates. When the cells were 50% confluent, the cells were treated with GNC-siRNA or Lipofectamine® 2000-transfected siRNA with different concentrations of siRNA. After 24 h incubation, the proportion of viable cells was evaluated using CCK-8 assay.

In the proliferation assay, Panc-1 cells were plated in 96-well plates. When the cells were 50% confluent, the cells were incubated with GNC-siRNA or GNC-nsRNA (100 nM siRNA). After 24 h, the cells were rinsed with PBS and incubated in fresh medium. Then, the proliferation of Panc-1 cells was assessed during another 72 h using CCK-8 assay.

Now, we have updated the details in the experimental section (Page 23 of the revised manuscript), and we have updated the time points in the X-axis in Fig. 4B for clarification (Page 36 of the revised manuscript).

Figure 4A&B

• Figure 4D,E. Are the authors looking at an effect on migration here or at an effect on proliferation?

Response: In fact, this is a migration assay of Panc-1 cells (Figure 3C). We previously developed this migration model based on microfluidic chips in our labs (PMID 17183592, PMID 19787666).

To definitely rule out an effect on proliferation, we redid this migration assay with pre-treatment of Panc-1 cells with mitotic inhibitor (mitomycin C) (Page 23, and Page 36 in the revised manuscript). The use of mitotic inhibitor inhibited the proliferation of Panc-1 cells, thus we can focus on the migration effect of the Panc-1 cells. However, after update of the experiment process, we obtained the similar tendency as previously that, the GNC-siRNA complex inhibited the migration of Panc-1 cells compared to nontreated cells and GNC-nsRNA treated group (updated Fig. 4D,E).

Figure 4

Reviewers' comments:

Reviewer #1 (Remarks to the Author):

In the revised version of the manuscript, I feel that the authors only partially addressed my concerns, which are listed as follows:

1. Sup. Fig1a: the difference in NGF tissue levels between normal pancreas and tumor looks very marginal.
2. Sup. Fig1b: the "normal" pancreatic tissue does still not appear normal. Please provide true, i.e. acinar, tissue images.
3. These data on Sup. Fig.1, i.e. the expression of NGF in human tissues, has been extensively reported in previous studies and does not add any novelty.
4. Sup. Fig 2.: Still, as I stated before, the biological studies should not only have been performed with one (i.e. the most potent) siRNA, but also with an additional, comparably strong siRNA for biological validity.
5. How does the elimination of GNC-siRNA take place? And how would the authors speculate on the low toxicity of GNC-siRNA as opposed to free siRNA?
6. The provided images on the Figure 9A, the right column, could compare the expression of necrosis and apoptosis markers in the treatment groups.

Reviewer #2 (Remarks to the Author):

The authors have adequately addressed all my comments.

Reviewer #1 (Remarks to the Author)

In the revised version of the manuscript, I feel that the authors only partially addressed my concerns, which are listed as follows:

Response: Thank you very much for reviewing our manuscript again and giving us the critical comments. All your comments help us to further improve the quality of our manuscript. Please find in the attached as a point-by-point response to your comments.

1. Sup. Fig1a: the difference in NGF tissue levels between normal pancreas and tumor looks very marginal.

Response: Thank you very much for pointing out this critical concern. We repeated the Western blot for four times, in order to show the difference of NGF protein level between pancreatic tumor tissues and normal pancreas tissues. As suggested, we quantified the optical density of the NGF bands. We defined the optical density of NGF bands in pancreatic tumor being 1.0, and the optical density of NGF in normal pancreas tissues was calculated to be 0.28 ± 0.08 ($p < 0.01$, $n = 4$). The statistical analysis indicated a significantly higher expression level of NGF in pancreatic tumor tissues than in normal pancreas tissues. We have updated this information in Page 4 of the revised manuscript and in Sup. Fig. 1A in Page 7 of the revised supporting information.

Moreover, the immunohistochemistry and immunofluorescence staining of NGF also revealed the same tendency, please refer to the response to the following question.

Sup. Fig. 1A

2. Sup. Fig1b: the “normal” pancreatic tissue does still not appear normal. Please provide true, i.e. acinar, tissue images.

Response: Thank you very much for pointing out this critical concern. As suggested, we used both immunohistochemistry (IHC) and immunofluorescence (IF) staining to evaluate the NGF expression between the normal pancreas and the pancreatic tumor tissues (Sup. Fig. 1B).

From the IHC images, we can clearly distinguish the acinar structure of pancreas tissue (Sup. Fig. 1B, left panel, the arrows in the IHC images represented the acinar structure of pancreas). Both IHC and IF staining revealed a higher NGF immunoreactivity in pancreatic tumors than in normal pancreas. We have updated the figure in Page 7 of the revised

supporting information.

Sup. Fig. 1B (Scale bars: 100 μ m)

3. These data on Sup. Fig.1, i.e. the expression of NGF in human tissues, has been extensively reported in previous studies and does not add any novelty.

Response: Thank you very much for pointing out this critical concern. In order for reader of interdisciplinary background to easily understand our subjects, we have provided these results. However, if both the Reviewer and the Editor have determined that we do not need these images in Sup. Fig. 1, we can get rid of them.

4. Sup. Fig 2.: Still, as I stated before, the biological studies should not only have been performed with one (i.e. the most potent) siRNA, but also with an additional, comparably strong siRNA for biological validity.

Response: Thank you very much for pointing out these critical concerns.

As suggested, within the 5 siRNA sequences initially used for the screening of the most efficient sequence (Sup. Fig. 2), we chose siRNA-#2 (the most potent), siRNA-#3 (the least potent) and siRNA-#5 (the middle potent) for more experiments as below.

Sup. Fig. 2

First, we evaluated the gene knockdown effect of various GNC-siRNA complexes in cellular level in Panc-1 cells (new Sup. Fig. 13). GNC-siRNA-#2 and GNC-siRNA-#5 both showed the knockdown of NGF expression (Sup. Fig. 13). And the GNC-siRNA-#2 (chosen in our previous study) showed the most effective knockdown of NGF (Sup. Fig.13). We have modified the manuscript accordingly (Page 7 of the revised manuscript, Page 19 of the revised supporting information).

Sup. Fig. 13

As suggested, we also conducted the study with various GNC-siRNA formulations in orthotopic pancreatic tumor models (new Sup. Fig. 16). GNC-siRNA-#2 and GNC-siRNA-#5 complexes showed more effective anti-tumor effects and NGF gene knockdown effects than GNC-siRNA-#3 in the orthotopic tumors (Sup. Fig. 16). We have updated the manuscript accordingly (Page 12 of the revised manuscript, Page 22 of the revised supporting information).

Sup. Fig. 16

5. How does the elimination of GNC-siRNA take place?

Response: Thank you very much for pointing out these critical concerns.

To explore the elimination mechanism of GNC-siRNA, we injected the GNC-siRNA complex into mice via systemic injection, and we used ICP-MS to measure the gold concentration in various organs, tumors (expressed as % of injected dose), blood and urine (expressed as ng g^{-1}) of mice at 1 h, 6 h, 12 h, 24 h and 48 h after injection. The results were shown as below (new Sup. Fig. 15):

Sup. Fig. 15

At 1 h after injection, the concentration of gold (from GNC-siRNA) in kidney and liver was the highest, followed by spleen, lung and heart.

As time goes on, the gold concentration in kidney decreased gradually, whereas the accumulation of gold in liver achieved the highest at 12 h and then decreased gradually. The gold accumulated at the tumor sites and achieved the highest concentration at 24 h, and then decreased gradually.

Thus, the GNC-siRNA is mainly eliminated by liver and kidney. It agrees with previous report that the liver can rapidly take up, degrade, inactivate and eliminate these nanomaterials (reference 1), while the kidney is an excretory organ where the nanoparticles could excreted into the urine (reference 2) (Sup. Fig. 15, right panel). We have updated the relevant information in Page 9-10 of the revised manuscript and in Page 21 of the revised supporting information.

And how would the authors speculate on the low toxicity of GNC-siRNA as opposed to free siRNA?

Response: From the literature, free siRNA duplexes are potent to activate the mammalian innate immune system, and lead to systemic inflammation *in vivo* through inducing high levels of inflammatory cytokines (references 3&4), thus the inflammatory responses by siRNA can result in significant toxicities *in vivo*. We used the GNC-siRNA complex to reduce the free siRNA-induced immunostimulation.

The reduced toxicity of GNC-siRNA compared to free siRNA was analyzed by both of blood biochemical parameters and immunotoxicity assay.

In the blood biochemical analysis, we found that GNC-siRNA complex kept the parameters of liver (ALT, AST) and kidney (BUN, CREA) functions within the normal ranges (Sup. Table 6). However, free siRNA induced a relatively high level of ALT and AST (Sup. Table 6), indicating hepatic dysfunction caused by the free siRNA treatment (reference 5).

Sup. Table 6. Biochemical parameters of blood from mice 24 h after injection of drugs

	ALT (U l ⁻¹)	AST (U l ⁻¹)	BUN (mM)	CREA (μM)
Saline	48 ± 11	132 ± 9	5.6 ± 0.6	32 ± 4
Free siRNA	154 ± 32	193 ± 37	7.2 ± 0.9	24 ± 3
GNC-siRNA	51 ± 12	168 ± 21	8.0 ± 0.8	41 ± 4
GNC-nsRNA	76 ± 21	160 ± 15	6.4 ± 0.9	25 ± 3
Reference	17-132	54-298	2.8-11.7	18-80

ALT, alanine amino transferase; AST, aspartate amino transferase; BUN, blood urea nitrogen; CREA, creatinine.

In the immunotoxicity assay, at the therapeutic doses, no significant immune response was induced by GNC-siRNA complex, as the production of serum cytokines, including

interleukin-6 (IL-6) and tumor necrosis factor- α (TNF- α) was similar to the saline control group (Sup. Fig. 18). However, the free siRNA group induced a relatively high immune responses by production of more IL-6 and TNF- α (Sup. Fig. 18) (reference 3).

Together, these results indicated that GNC-siRNA complex showed high safety and low toxicity in vivo compared to free siRNA. We have updated this information in Page 13-14 of the revised manuscript and in Page 24 of the revised supporting information.

Sup. Fig. 18

6. The provided images on the Figure 9A (note: now Sup. Fig. 17A in this revised version), the right column, could compare the expression of necrosis and apoptosis markers in the treatment groups.

Response: Thank you very much for pointing out this critical concern.

As suggested, we used the TUNEL kit (Roche, 12156792910) to detect the apoptosis in the tumor sections, since TUNEL (Terminal deoxynucleotidyl transferase dUTP nick end labeling) is a common method for detecting DNA fragmentation that from apoptotic signaling cascade. In this assay, cells in the tumor tissues undergoing apoptosis (TUNEL-positive cells) were labeled red, whereas the cell nuclei were labeled by DAPI in blue (Sup. Fig. 17B).

And we evaluated the necrosis markers using a modified necrosis detection kit (Abcam, 176749). The necrosis was labeled by 7-aminoactinomycin D (7-AAD, a dye labels the nucleus of damaged cells in necrosis, ex/em = 550/650 nm, red fluorescence), and the cell nuclei were stained with DAPI (blue fluorescence), respectively (Sup. Fig. 17C).

And we quantified the apoptosis and necrosis level as the percentage of TUNEL-positive or 7-AAD-positive cells in the slices, respectively (Sup. Fig. 17D). The results showed that the GNC-siRNA treatment induced a higher apoptosis and necrosis level in the tumor tissues (Sup. Fig. 17B-D). We have updated the relevant information accordingly (Page 13 of the revised manuscript, Page 5-6 and Page 23 of the supporting information).

Sup. Fig. 17

References

1. Sadauskas, E. et al. Kupffer cells are central in the removal of nanoparticles from the organism. *Part. Fibre Toxicol.* **4**, 10 (2007).
2. Soo Choi, H. et al. Renal clearance of quantum dots. *Nat. Biotechnol.* **25**, 1165-1170 (2007).
3. Judge, A. D. et al. Sequence-dependent stimulation of the mammalian innate immune response by synthetic siRNA. *Nat. Biotechnol.* **23**, 457-462 (2005).
4. Sioud, M. Induction of inflammatory cytokines and interferon responses by double-stranded and single-stranded siRNAs is sequence-dependent and requires endosomal localization. *J. Mol. Biol.* **348**, 1079-1090 (2005).
5. Wang, J. et al. Acute toxicity and biodistribution of different sized titanium dioxide particles in mice after oral administration. *Toxicol. Lett.* **168**, 176-185 (2007).

REVIEWERS' COMMENTS:

Reviewer #1 (Remarks to the Author):

In the re-revised version of their manuscript, the authors have satisfyingly addressed my concerns. Their results look plausible and concordant.

My remaining concern, as also in the first version of the manuscript, is that, I feel any relevant pharmacological study on pancreatic cancer should make use of genetic models of pancreatic cancer that harbor the desmoplasia that "defines" pancreatic cancer and affects its most important pharmacological features. The models included in this study, including s.c. injection, PDX or orthotopic injections, do not reliably reproduce this extent of desmoplasia, and thus translatability is most probably very limited.

Reviewer #1 (Remarks to the Author)

In the re-revised version of their manuscript, the authors have satisfyingly addressed my concerns. Their results look plausible and concordant.

My remaining concern, as also in the first version of the manuscript, is that, I feel any relevant pharmacological study on pancreatic cancer should make use of genetic models of pancreatic cancer that harbor the desmoplasia that “defines” pancreatic cancer and affects its most important pharmacological features. The models included in this study, including s.c. injection, PDX or orthotopic injections, do not reliably reproduce this extent of desmoplasia, and thus translatability is most probably very limited.

Response: Thank you very much for your re-reviewing our manuscript and giving us your critical comments. Your suggestions are helpful for further improving the quality of our manuscript.

The genetic models of pancreatic cancer are critical for translation medicine and we will be willing to do this in our following research.

We notice that a hallmark feature of pancreatic cancer is the dense desmoplasia of the pancreatic tumors, which may hamper the perfusion and uptake of drugs. We thank the reviewer for the suggestion to use genetically engineered mouse models (such as KPC model) for pancreatic cancer. It is known KPC model is a well-validated, clinically relevant model of pancreatic cancer,^{1,2} and the tumors in KPC models generally develop extensive desmoplasia.^{1,3}

However, as mentioned previously, the KPC model mice are not available in China presently, and the cross-border transport of the model animals from the relevant labs is difficult. Although we are very interested in testing the efficiency of GNC-siRNA complex on the KPC model, we are sorry to have difficulty to carry out this assay in the present study.

In this study, we have used the patient-derived xenograft (PDX) pancreatic tumor model. PDX pancreatic models employ the implantation of primary human pancreatic cancer specimens as orthotopic models.^{4,5} These PDX pancreatic models are characterized by the maintenance of the original tumor architecture. Although the human stroma is replaced by murine stroma, orthotopic PDX pancreatic models retain a greater proportion of stromal components and develop metastasis.^{4,5} Thus the PDX models offer a great *in vivo* platform for translational cancer research and drug development in pancreatic cancer.^{6,7}

Our present study (using subcutaneous, orthotopic and PDX tumor models) is in an initial pharmacological exploratory phase. Although the results in this study are far from the clinical translations yet, we believe that our study may provide useful guidance to the treatment and drug development of pancreatic cancer.

As suggested, we are trying to establish the genetic mouse models and will try to use the genetic mice in our future studies, and we have added a discussion on the perspectives

of this study using genetic mouse models of pancreatic cancer (Pages 15-16 of the revised manuscript). The added part is shown below:

“The genetically engineered mouse models of pancreatic cancer, such as the *LSL-Kras^{G12D/+}*; *LSL-Trp53^{R172H/+}*; *Pdx-1-Cre* (KPC) mice,^{1,2} can generate the dense desmoplasia which defines pancreatic cancer and affects its most important pharmacological features.^{1,3} In the future study, we will use the genetically engineered mouse models of pancreatic cancer to explore the translatability of the GNC-siRNA complex.”

References

- 1 Hingorani, S. R. *et al.* Trp53R172H and KrasG12D cooperate to promote chromosomal instability and widely metastatic pancreatic ductal adenocarcinoma in mice. *Cancer Cell* **7**, 469-483 (2005).
- 2 Olive, K. P. *et al.* Inhibition of Hedgehog signaling enhances delivery of chemotherapy in a mouse model of pancreatic cancer. *Science* **324**, 1457-1461 (2009).
- 3 Gopinathan, A., Morton, J. P., Jodrell, D. I. & Sansom, O. J. GEMMs as preclinical models for testing pancreatic cancer therapies. *Dis. Mod. Mech.* **8**, 1185-1200 (2015).
- 4 Kim, M. P. *et al.* Generation of orthotopic and heterotopic human pancreatic cancer xenografts in immunodeficient mice. *Nat. Protoc.* **4**, 1670-1680 (2009).
- 5 Fu, X., Guadagni, F. & Hoffman, R. M. A metastatic nude-mouse model of human pancreatic cancer constructed orthotopically with histologically intact patient specimens. *Proc. Natl. Acad. Sci. USA* **89**, 5645-5649 (1992).
- 6 Tentler, J. J. *et al.* Patient-derived tumour xenografts as models for oncology drug development. *Nat. Rev. Clin. Oncol.* **9**, 338-350 (2012).
- 7 Siolas, D. & Hannon, G. J. Patient-derived tumor xenografts: transforming clinical samples into mouse models. *Cancer Res.* **73**, 5315-5319 (2013).